# Endogenous opioid receptor system mediates costly altruism in the human brain
Jinglu Chen [1] ✉, Vesa Putkinen[1,2], Kerttu Seppälä [1,3], Jussi Hirvonen[1,4], Kalliopi Ioumpa[5], Valeria Gazzola [5,6], Christian Keysers [5,6] & Lauri Nummenmaa [1,7,8]

Functional neuroimaging studies suggest that a large-scale brain network transforms others' pain into its vicarious representation in the observer, potentially modulating helping behavior. However, the neuromolecular basis of individual differences in vicarious pain and helping is poorly understood. We investigated the role of the endogenous μ-opioid receptor (MOR) system in altruistic costly helping. MOR density was measured using [$^{11}$C]carfentanil. In a separate fMRI experiment, participants could donate money to reduce a confederate's pain from electric shocks. Participants were generally willing to help, and brain activity was observed in amygdala, anterior insula, anterior cingulate cortex (ACC), striatum, primary motor cortex, primary somatosensory cortex and thalamus when witnessing others' pain. Haemodynamic responses were negatively associated with MOR availability in emotion circuits. However, MOR availability positively associated with the ACC and hippocampus during helping. These findings suggest that the endogenous MOR system modulates altruism in the human brain.

Prosocial behavior is prevalent in humans and animals. Social animals share emotions with each other, comfort peer's distress, and help others in times of need[1,2]. Humans help each other in daily life even without genetic relatedness or obvious direct profit[3–5]. Such altruistic behavior by definition benefits the receiver, but also has a range of positive outcomes for the helper, including improved academic performance and social preferences[6], increased acceptance by peers at school[7], and higher life satisfaction[8]. For more information, see review ref. 9. The evolutionary routes of altruism are a topic of enduring interest[10]. A prominent hypothesis for the proximate causes for altruistic helping holds that if we empathically activate our own pain when witnessing the pain of others, mechanisms that have evolved to motivate us to prevent damage to ourselves will also motivate us to prevent pain and damage to others. Accordingly, empathy can be hypothesized to drive prosociality[11–14].

The neurocognitive link between empathy and prosociality has been investigated by quantifying responses in brain regions associated with empathy while witnessing the pain of others, and correlating these measures with individual differences in helping. Studies leveraging a range of different methods have consistently shown that seeing or hearing others in distress compared to neutral states engages a core network including the anterior insula (aIns) and adjacent frontal operculum, the mid- to anterior- cingulate

cortex, amygdala and, when the somatic source of the pain is salient, the primary and secondary somatosensory cortices (SI and SII)[1,15–19]. Based on the details of the task and the nature of how the pain is witnessed, this extends into a larger network also encompassing primary motor cortex (M1), extensive areas in frontal lobe and parietal lobe, dorsolateral prefrontal cortex (dlPFC), ventromedial frontal cortex (vmPFC), fusiform gyrus, temporal pole, precuneus, thalamus, caudate, and putamen[20,21]. Particularly the anterior cingulate cortex (ACC) and aIns are activated both when subjects experience first-hand pain and when they see others experiencing pain, suggesting that these regions underlie the transformation of others' pain from visual and auditory inputs into sensorimotor formats[15,17,19,22–24]. The ACC has been shown to contain single neurons that respond to both witnessing and experiencing pain in rodents[25].

To investigate whether this recruitment of regions involved in the witnesses' own pain plays a role in motivating helping, some studies have gone beyond simply measuring brain activity while passively witnessing the pain of others and provided the participant with an opportunity to help. One study offered participants the opportunity to relieve other people's pain by taking some of that pain onto themselves and found that anterior insular activity predicted the willingness to help ingroup members and nucleus accumbens activity predicted reluctance to help outgroup members[26].

[1]Turku PET Centre, Turku University Hospital and University of Turku, Turku, Finland. [2]Turku Institute for Advanced Studies, University of Turku, Turku, Finland. [3]Department of Medical Physics, Turku University Hospital, Turku, Finland. [4]Department of Radiology, Turku University Hospital, Turku, Finland. [5]The Netherlands Institute for Neuroscience, KNAW research institute, Amsterdam, The Netherlands. [6]Department of Psychology, University of Amsterdam, Amsterdam, The Netherlands. [7]Turku University Hospital, Turku, Finland. [8]Department of Psychology, University of Turku, Turku, Finland. ✉e-mail: jinglu.chen@utu.fi

Another found that reduced functional connectivity between the insula and ACC characterized participants that decide to help a person in need in a virtual reality scenario[27]. Activity in somatosensory and insular region while witnessing pain predicted later charitable donations, albeit in ways that depended on socioeconomic status[28]. An approach that has been particularly successful in identifying brain regions with activity quantitatively associated with helping, involves offering participants a certain financial endowment, and enabling them to forfeit some of that endowment on a trial-by-trial basis to reduce the pain of a victim in an immersive experimental setting[29]. These studies have found a variety of nodes also involved with self-pain to relate to individual or trial-by-trial differences in helping, including the dlPFC, orbital frontal cortex[30], SI[31], vmPFC[32,33], insula, ACC, and amygdala[34].

Although these financially costly helping paradigms thus start to provide some traction on which neural networks have BOLD activity that correlate with costly helping, the molecular basis of empathy and costly helping, and their individual differences, remain poorly understood. Multiple lines of evidence point towards the critical role of the endogenous opioid system and particularly the endogenous μ-opioid receptor (MOR) system in the first-person experience of pain and in differences in empathy and prosociality. Among the three classes of opioid receptors (μ, δ, and κ), the μ receptors mediate the effects of endogenous β-endorphins, endomorphins, enkephalins, and various exogenous opioid agonists[35]. The predominant action of μ-opioids in the central nervous system is inhibitory, but they can also exert excitatory effects, and MORs are expressed widely throughout the human emotion circuits[36,37]. Opioids are well known for their role in antinociception, and genetic deletions of the MOR in mice abolishes the analgesic effect of opioid agonists[38], demonstrating that this receptor is the sole pathway for the analgesic effects of these drugs. The effect of opioids however goes beyond nociception: opioid agonists decrease and antagonists increase social motivation in macaques[39,40], and in humans, the opioid system modulates both positive and negative emotions[37,41].

With regard to empathy, recent experiments have shown that activating the MOR system using placebo analgesia reduces first-hand pain ratings, how much pain participants perceive in others, and how unpleasant they find witnessing such pain[42,43], and this effect can be blocked using a MOR antagonist[42]. Also, the long term use of opioids, known to dysregulate the MOR system, leads to reduced pain ratings in others[44]. One study also showed that placebo analgesia reduces helping[45]. Positron emission tomography (PET) with [11C] carfentanil, a synthetic, highly specific MOR agonist allows in vivo imaging of opioid receptors in humans. Studies with PET have demonstrated endogenous opioid release in the thalamus following acute pain in a dose-dependent manner[46]. Individual differences in MOR availability also link with pain sensitivity: participants with lower MOR availability have a higher sensitivity to pain[47]. Finally, and critically with respect to the present study, PET studies have linked opioid receptor availability with vicarious pain perception and sociability. First, MOR availability is negatively associated with haemodynamic responses to seeing others in pain[48]. The MOR system is also activated during positive social interactions such as laughing together[49], and MOR availability is positively correlated with prosocial motivation as indexed by social attachment styles[50,51]. Against this background, it could be hypothesized that the MOR system would be a crucial molecular pathway for altruistic, costly helping, but currently, this hypothesis lacks direct in vivo evidence.

## The current study

Here we investigated whether individual differences in the MOR availability at rest translate into measurable differences in the willingness to forfeit money to reduce pain to others, using the established "costly" helping paradigm of Gallo et al. In the fMRI experiment, participants could choose to donate money to reduce the pain of the confederate who was subjected to electric shocks of varying intensity. In a separate scanning session, the participants underwent a baseline PET scan with the MOR specific radioligand [11C]carfentanil. We expected that participants would show a propensity to donate money to alleviate the pain of the confederate, as suggested by prior research[52]. Additionally, previous studies suggest a potential negative correlation between MOR availability and brain regions associated with empathy and pain[48]. Therefore, we hypothesized a negative correlation between MOR availability and viewing others' pain. Finally, we hypothesized that individuals with high MOR availability will demonstrate greater altruistic tendencies at both behavioral and neural levels, given the positive correlation between MOR availability and prosocial motivation[50,51]. We found that people were in general willing to engage in costly helping; The fMRI results revealed that activity in amydala, aIns, ACC, striatum, primary motor cortex, primary somatosensory cortex, and thalamus increased when participants saw the confederate in pain. These haemodynamic responses had amplitudes that differed across individuals in ways that correlated with the availability of MORs in the striatolimbic and cortical emotion circuits. Altogether these data suggest that endogenous MOR system contributes to altruistic brain and its individual differences.

## Materials and methods
### Participants

Thirty healthy Finnish women (mean ± SD age: 24.7 ± 5.65 years, range 19–42) with normal or corrected to normal vision were recruited in the study. To maximize statistical power of the study, only females were included because the MOR system shows a high level of sexual dimorphism[36]. In addition, research on sex differences in empathy and pain shows that women are better at judging emotional signals[53], show higher emotional responsivity[54], evaluate others' pain as more intense[55], are more empathic than men[56], and their altruistic behavior are not influenced by trait harm sensitivity[57]. All subjects participated in the fMRI scan and 14 of them participated in the PET scan. PET and MRI scans were conducted on separate days. Exclusion criteria included medications affecting the central nervous system, mood or anxiety disorders, psychotic disorders or neurological conditions, substance abuse, and standard MRI and PET exclusion criteria. Structural brain abnormalities that are clinically relevant or could bias the analyses were excluded by a consultant neuroradiologist. The study was approved by the ethics board of the hospital district of Southwest Finland and conducted according to Good Clinical Practice and the Declaration of Helsinki (Approval number: 51/1801/2019). All subjects signed written informed consent and were informed that they had the right to withdraw at any time without giving a reason. The participants were compensated for their time and travel costs.

### Experimental design and stimuli for fMRI

The fMRI study was run using the costly helping paradigm of ref. 31. The participants were led to believe that they were witnessing the distress of another person in real time. Before the experiment, participants met the confederate (author KS) and were explained that the experiment would be performed in two separate rooms connected by a video camera. They were invited to draw lots to determine who would undergo the fMRI measurement while seeing the other participant receiving the electric shocks. The confederate was always chosen to receive the electric shocks. During the scan, participants, believing to witness the pain of the victim through a closed-circuit television, actually they viewed pre-recorded videos of the confederate receiving the painful simulation. All participants were debriefed at the end of the experiment.

The task was run by Presentation software (https://www.neurobs.com/) and the trial structure of the experiment is shown in Fig. 1. On each trial, subjects first saw a video of the confederate receiving a painful electric shock (1st video). The intensity of expressed pain ranged randomly from 2 (mild pain) to 6 (moderate pain) out of ten. After the video, subjects could decide how much money they were willing to donate on that trial to reduce the intensity of the second shock in that trial, to do that, one button was held by the participant in each hand, one representing increasing money and the other representing decreasing money. For each trial, they were given 6€. They knew that if they did not donate any money, the second shock would have the same intensity as the first, whilst, for every donated 1€, the second shock would be reduced by one point on the 10-point pain scale. Participants were

told that they can keep the undonated money of all trials divided by 10 as their extra compensation after the experiment. The subjects then saw the 2nd video of the confederate receiving the second electric shock whose intensity reflected the first shock minus the donation (see supplementary for more information on videos, Supplementary Fig. S1). There were two imaging runs with 15 trials in each.

## MRI data acquisition and preprocessing
fMRI data were acquired with the 3T scanner (SIGNA, Premier, GE Healthcare, Waukesha, WI, USA) at the University Hospital of Turku. T2*-weighted functional images were collected with echo-planner imaging (EPI) sequence (45 slices; slice thickness = 3 mm; TR = 2600 ms; TE = 30 ms; flip angle =75°; FOV = 24 mm; voxel size = 3 × 3 × 3 mm³). T1-weighted structural images were collected with voxel size of 1 × 1 × 1 mm³. MRI data were preprocessed with fMRIPrep 1.3.0.post2[58]. The following pre-processing was performed on the T1-weighted (T1w) image: correction for intensity, skull-stripping, brain surfaces reconstruction, refined brain mask estimating, cortical gray-matter segmentations, spatial normalization to the ICBM 152 Nonlinear Asymmetrical template version 2009c using nonlinear registration with antsRegistration (ANTs 2.2.0), and brain tissue segmentation. The following preprocessing was performed on the functional data: generation of reference volume and its skull-stripped version, co-registration to the T1w reference, slice-time correction, spatial smoothing with an isotropic, Gaussian Kernel of 6 mm FWHM (full-width half-maximum), automatic removal of motion artifacts using ICA-AROMA[59], and resampling to MNI152NLin2009cAsym standard space, and principal components are estimated after high-pass filtering the preprocessed BOLD time-series for the two CompCor variants: temporal and anatomical.

## PET data acquisition and analysis
PET data were acquired during resting baseline with a GE Discovery Molecular Insights DMI PET/CT, GE Healthcare, Waukesha WI in Turku PET Center. The high-affinity agonist radioligand [¹¹C]carfentanil was used to measure brain μ-opioid receptor availability. After intravenous radioligand injection (250.6 ± 10.9 MBq), radioactivity in the brain was measured by the PET scanner for 51 min with increasing frame length (3 × 60 s, 4 × 180 s, 6 × 360 s) using in-plane resolution of 3.77 mm FWHM (Full Width Half Maximum) and tangential 4.00 mm FWHM. All subjects lay supine in the PET scanner throughout the study. Data were corrected for dead-time, decay, and measured photon attenuation. In-house MAGIA pipeline was used to preprocess PET images[60].

Radiotracer binding was quantified using binding potential ($BP_{ND}$), calculated as the ratio of specific binding to non-displaceable binding in the tissue[61]. This outcome measure is not confounded by differences in peripheral distribution or radiotracer metabolism, or alterations in brain perfusion. $BP_{ND}$ is traditionally interpreted by target molecule density ($B_{max}$), although [¹¹C]carfentanil is also sensitive to endogenous neurotransmitter release. Accordingly, the $BP_{ND}$ for the tracer should be interpreted as the density of the receptors unoccupied by endogenous ligand (i.e., receptor availability). Binding potential was calculated by applying the basis function method[62] for each voxel using the simplified reference tissue model[63], with

occipital cortex serving as the reference region[64]. The parametric images were spatially normalized to MNI-space via segmentation and normalization of T1-weighted anatomical images, and finally smoothed with an 8-mm FWHM Gaussian kernel. PET imaging with [¹¹C]carfentanil has high test-retest stability[65]. PET imaging always preceded fMRI to avoid potential impact of the fMRI tasks on measured MOR levels (mean ± SD day: 84 ± 62 days, range 4–190).

## ROI selection
The average tracer $BP_{ND}$ was quantified in 17 anatomical a priori regions of interest (ROI) involved in vicarious pain and empathy: amygdala, caudate, cerebellum, dorsal ACC, inferior temporal gyrus, insula, middle temporal gyrus, nucleus accumbens, orbitofrontal cortex, pars opercularis, posterior cingulate cortex, putamen, rostral ACC, superior frontal gyrus, superior temporal gyrus, temporal pole, and thalamus. The selection was based on previous studies on the effects of MORs on vicarious pain and arousal[48,66]. The ROIs were derived separately for each subject from the T1-weighted MR images using FreeSurfer (http://surfer.nmr.mgh.harvard.edu/).

## Statistics and reproducibility
fMRI data were analyzed in SPM12 (Wellcome Trust Center for Imaging, London, UK, https://www.fil.ion.ucl.ac.uk/spm/) ($N = 30$). To investigate regions activated by (i) seeing pain in the 1st video, (ii) donating, and (iii) seeing pain in the 2nd video, first-level general linear models (GLM) were estimated by modeling the 1st video, donation phase, and the 2nd video by using boxcar regressors convolved with HRF in the design matrix. Donation size (trial-wise donations for each subject) was entered as parametric modulator for the 1st video. Subject-wise contrast images were then generated for main effects of 1st video, donation phase, 2nd video. Additionally, a subtraction contrast was computed for 1st vs. 2nd video. The contrast images were then subjected to second-level (random effects) analysis. Results are shown after FWE correction for cluster-size, by initially thresholding statistical maps at $p_{unc} < 0.001$, identifying the FWEc minimum cluster-size value for FWE correction at the cluster-size level, and then thresholding the statistical maps again at $p_{unc} < 0.001$ and k=FWEc.

Three approaches were taken to characterize the interactions between MOR availability and BOLD responses in pain perception and costly altruism ($N = 14$). In the first two approaches, a principal component analysis (PCA) was used to reduce the dimensionality in the $BP_{ND}$ values across our ROIs. This was done because regional MOR availability has high autocorrelation[67], thus PCA would increase the power of our analyses by reducing the multiple comparison correction that would otherwise reduce power. We found the first 3 PCs to explain >90% of the variance, with 61%, 22% and 7% of variance explained, respectively. To identify voxels with BOLD responses that depend on individual differences in the overall MOR signal, we used the first PC to predict the voxel-wise BOLD responses to the 1st video with donation size as a parametric modulator and the donation phase, separately. Specifically, we used the same parametric model as the fMRI analysis in the first-level model, and input the first PC in the second-level model for 1st video and donation, in separate models. Second, to explore the relationship between individual MOR differences and responses

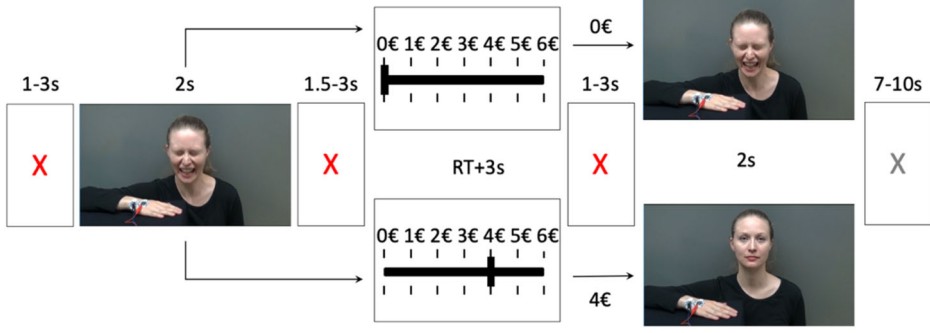

**Fig. 1 | Trial structure.** A red fixation cross was shown for 1–3 s and was followed by 1st video for 2 s. Then another red fixation cross was shown for 1.5–3 s. The donation phase was self-paced and was followed by a red fixation cross (1–3 s). Next, the post-donation video was shown for 2 s followed by a gray fixation cross (7–10 s). The snapshots from the videos shown on the right illustrate two possible scenarios from the 6 possibilities.

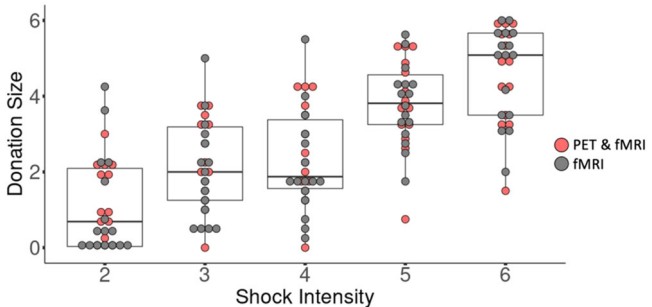

**Fig. 2 | Donation size increased with increasing shock intensity.** Each dot represents the average donation of a participant for all the stimuli of that intensity. *N* = 30 participants. The box represents the 1st, 2nd, and 3rd quartile, the whiskers the highest and lowest value within 1.5 interquartile range. Red dots represent participants who undertook both PET and fMRI scans. Gray dots represent participants with only fMRI scans.

**Table 1 | Lack of association between individual differences in slope or intercept and MOR PCs**

| donation | MOR | Pearson's *r* | *p* | BF$_{10}$ |
|---|---|---|---|---|
| slope | pca1 | 0.075 | 0.800 | 0.339 |
| slope | pca2 | 0.013 | 0.965 | 0.329 |
| slope | pca3 | −0.070 | 0.813 | 0.338 |
| intercept | pca1 | −0.100 | 0.734 | 0.347 |
| intercept | pca2 | −0.136 | 0.642 | 0.363 |
| intercept | pca3 | 0.337 | 0.239 | 0.621 |

For each of the 14 PET participants, we estimated the slope and intercept of the regression donation = slope*intensity + intercept. We then calculated the Pearson's r value between the 14 slope (top) or intercept (bottom rows) with Jasp (https://jasp-stats.org/).

in the vicarious pain observation network, we used the affective vicarious pain signature (AVPS) to dot-multiply the 1st level beta maps to 1st video with donation size as a parametric regressor for each participant[68], thereby reducing each parameter estimate volume to a scalar value, and computed the correlation between the resulting value and the first 3 MOR PCs. Finally, to replicate previous studies on the links between MOR availability and haemodynamic responses to vicarious pain and arousal[48,66], the voxel-wise BOLD responses to donation size in 1st video were predicted with ROI-wise [$^{11}$C]carfentanil binding potentials using whole-brain linear regression analysis with a statistical threshold set at *p* < 0.05, FWE-corrected at cluster-level. We then computed a cumulative map of the binarized MOR × BOLD beta maps to highlight regions whose BOLD responses were most consistently dependent on regional MOR availability.

### Reporting summary
Further information on research design is available in the Nature Portfolio Reporting Summary linked to this article.

## Results
### Behavioral results
Participants donated money in all intensity conditions (*M* = 2.826, SD = 1.964), and a linear mixed model confirmed that donations increased as a function of the shock intensity shown in the first video, *β* = 0.877, *SE* = 0.062, *t* = 14.049, *p* < 0.001, intercept is 0.196 (Fig. 2). The slope (beta = 0.877) indicates that participants overall donate money to reduce shock intensity by 88%, adapting their donations very precisely to how much pain was at stake. In addition, in the linear mixed model, we added subject as a random effect. For the intercept, variance is 1.943 and SD is 1.394. For the slope, variance is 0.104, and SD is 0.323. Given that the sensitivity of reward from helping behavior may vary over time, we included time as a factor in our model. The results can be found in the Supplementary Material (Temporal influence on donation behavior, Supplementary Fig. S9).

To explore whether donations depended on the MOR availability for the 14 participants who also underwent the PET scan, for each participant, we calculated the slope and intercept linking the participants donation with the intensity of shocks shown in 1st video (donation = slope*intensity + intercept). We then explored whether individual differences in slope or intercept were associated with individual differences in the MOR availability using the first 3 principal components of the MOR. We first did so using a multiple regression including all three PCs, which did not yield significant results for the slope (F(3,10) = 0.036, *p* = 0.990) or intercept (F(3,10) = 0.552, *p* = 0.658). Performing Bayesian correlation tests between the slope and intercept and each individual PCA (Table 1), confirms that this data is more likely under null hypothesis of no association. Together, this suggest that the actual donations do not depend on MOR availability in our ROIs. We therefore explored what brain circuits are involved in making

the donation decisions, and whether these circuits vary based on MOR availability even if the outcomes of the decisions do not.

### BOLD-fMRI responses to confederate's initial pain
Full random effects results maps are available on NeuroVault (https://identifiers.org/neurovault.collection:14151). Consistent with previous studies, whole-brain GLM revealed that aIns, ACC, and thalamus were activated when seeing the confederate in pain in the 1st video (Fig. 3). Additional activations were observed in the amygdala and striatum, which are key nodes of the emotion and reward networks. We next modeled the BOLD signal with the donation size in the 1st video as a regressor. Consistent with previous research, insula, ACC activated more strongly as the donation increased (Fig. 4)[19,22–24,48,52,69,70]. To discern whether the observed effect stems from witnessing others in pain rather than viewing videos, we utilized shock intensity as a regressor instead of donation size (Supplementary Fig. S8). Further details can be found in the Supplementary Materials.

### BOLD-fMRI response changes following costly helping
In each trial, participants viewed two videos, one before their decision, and another following their decision. Comparing neural responses to the 1st and 2nd videos revealed decreased responses to the 2nd video in areas involved in vicarious pain perception such as, insula, thalamus and ACC. Additional effects were found in striatum, M1 and S1 (Fig. 5). Because subjects were on average willing to help, the 2nd video condition contained predominantly low shock intensity clips, thus direct comparison of 1st and 2nd could simply reflect differences in mean pain intensity across conditions. We subsequently restricted this analysis to low shock intensity levels (level 2 and level 3) in the 1st video. Even when pain intensity was thus approximately matched across the 1st and 2nd videos (Fig. 6), effects were similar to those in Fig. 5, with thalamus, striatum, ACC, M1, and S1 showing decreased responses to the 2nd video (Fig. 6).

One intriguing finding is the apparent loss of activation in the insula. As illustrated in Fig. 4, there is heightened activation of the insula in response to larger donations, indicating its sensitivity to donation size. Therefore, it is conceivable that insula activation remains consistent between the low-shock intensity in the first video and the second video, given that the latter primarily features content with low shock intensity.

### MOR-dependent responses to empathy for pain
To test our hypothesis that individual differences in MOR availability would be associated with differences in the neural circuitry associated with taking helping decisions, we next modeled BOLD responses associated with donation size during the 1st video using regional MOR availabilities as predictors. We first used a PCA that captures the individual variability across the ROIs and extracted the first principal component that explained 60.7% of the individual variation in MOR binding (see Supplementary Table S1 for details). The component scores were used as regressors to predict the haemodynamic response to donation size during the 1st video.

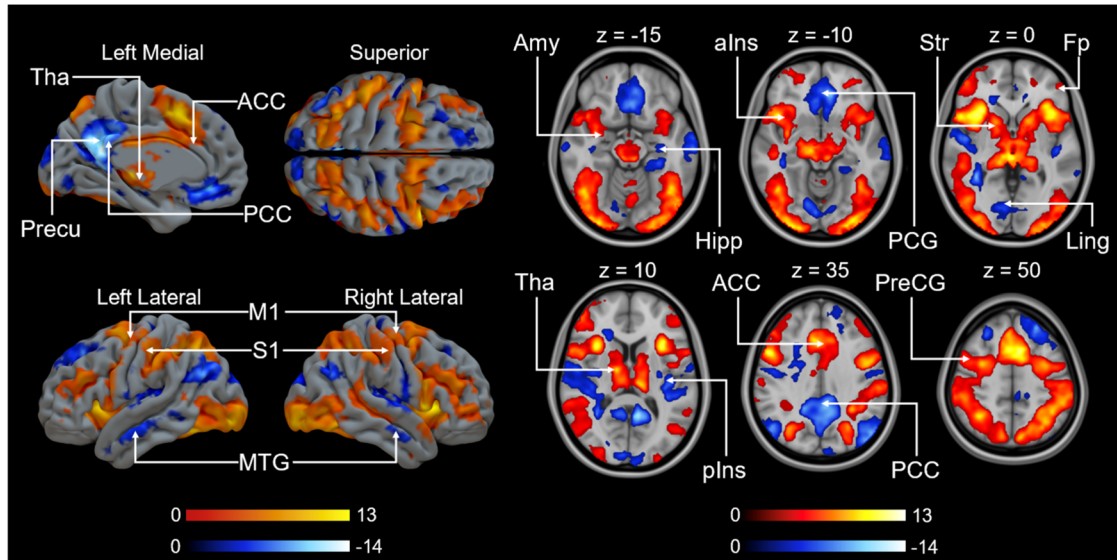

**Fig. 3 | Main effect of brain responses to the first video.** The data are thresholded at $p < 0.001$, FWE corrected at the cluster level (Positive: $p_{unc} < 0.001$, $k$ = FWEc = 114 voxels, $3.40 < t < 13.17$; Negative: $p_{unc} < 0.001$, $k$ = FWEc = 97 voxels, $3.40 < t < 15.77$). Colourbars indicate $t$ statistic range. Tha thalamus, Precu precuneus cortex, ACC anterior cingulate cortex, PCC posterior cingulate cortex, M1 primary motor cortex, S1 primary somatosensory cortex, MTG middle temporal gyrus, Amy amygdala, Hipp hippocampus, aIns anterior insula, PCG paracingulate gyrus, Str striatum, Fp frontal pole, Ling lingual gyrus, pIns posterior insula, PreCG precentral gyrus.

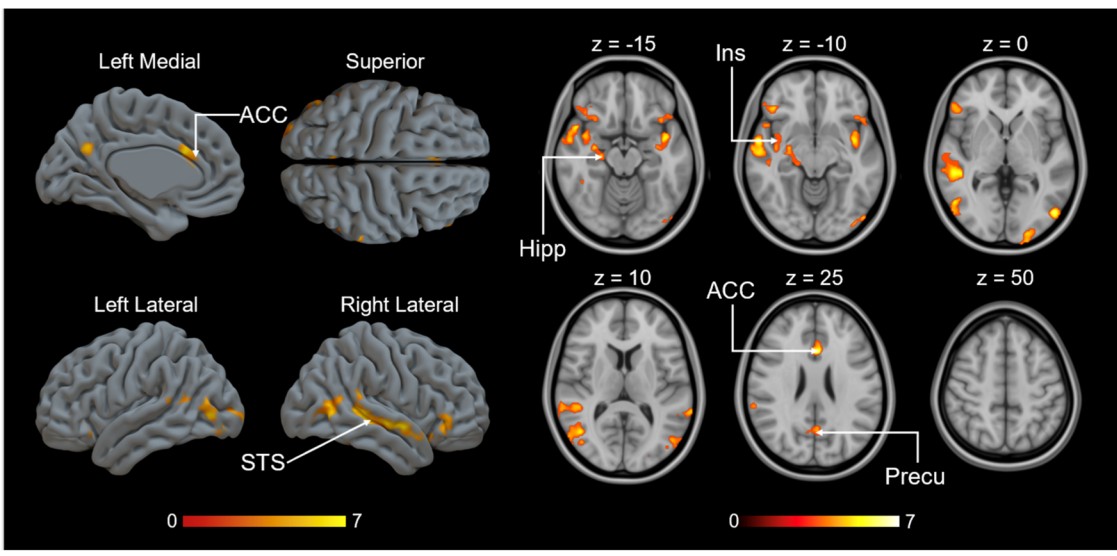

**Fig. 4 | Brain regions with BOLD signals associated with donation size during the 1st video.** Colourbars indicate $t$ statistic range. The data are thresholded at $p < 0.001$, FWE corrected at the cluster level ($p_{unc} < 0.001$, $k$ = FWEc = 163 voxels, $3.40 < t < 7.20$). ACC anterior cingulate cortex, Hipp hippocampus, Precu precuneus cortex, STS superior temporal sulcus, Ins insula.

We found a generally negative correlation between BOLD signal and MOR availability, mainly in amygdala, striatum, insula, hippocampus, thalamus, ACC and posterior cingulate cortex (Fig. 7). This indicates that participants with reduced MOR availability show BOLD signals in these regions that are more sensitive to trial-by-trial differences in donation. Importantly, these differences in the association between brain activity and donation as a function of MOR availability were observed despite similar donations across participants with higher or lower MOR availability (Table 1), ascertaining that these differences are not related to systematic differences in the predictors inserted into the first level fMRI models (see ref. 71 for a discussion of the importance of this factor).

To focus more specifically on regions associated with witnessing the pain of others, we used the AVPS signature, which was found to be negatively associated with the third principal component ($r = -0.659$, $p = 0.010$, $BF_{10} = 6.5$) (Supplementary Fig. S5). We also generated a cumulative map of correlation between regional MOR availabilities and BOLD responses to donation size in the 1st video. The results replicated our prior study revealing a generally negative correlation between BOLD signal and MOR availability in vicarious pain related areas like the insula, ACC, and thalamus. Extensive association between MOR availability and brain activation was also observed in somatosensory areas, temporal gyrus, limbic regions, and frontal cortex (Supplementary Fig. S2)[48].

**MOR-dependent responses to costly helping**

Finally, we explored whether individual differences in MOR availability were associated with the BOLD activity during the donation phase

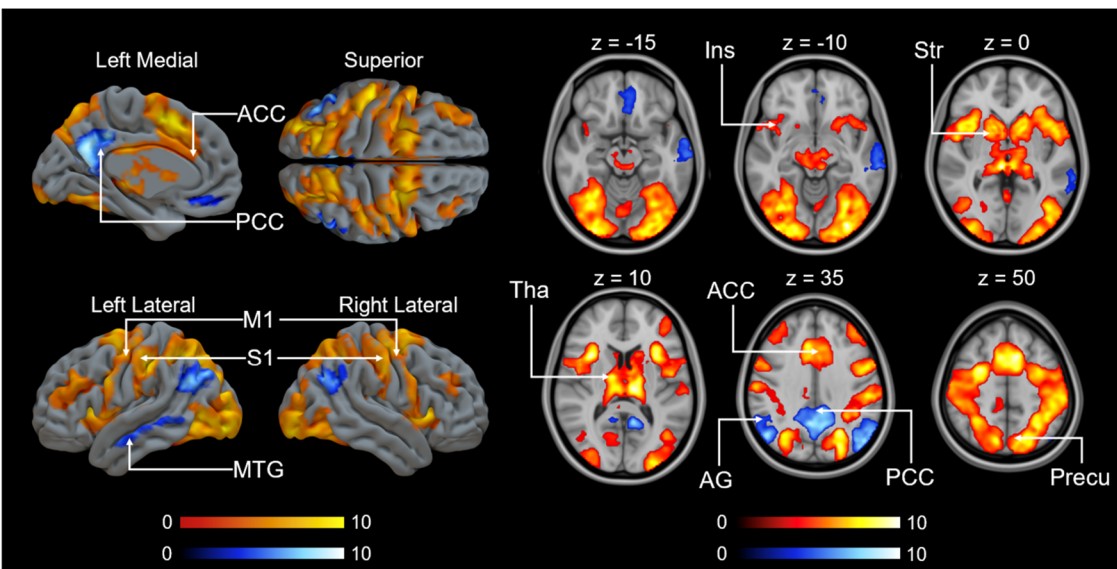

**Fig. 5 | Brain activation for the 1st versus the 2nd video.** The data are thresholded at $p < 0.001$, FWE corrected at the cluster level ($1^{st} > 2^{nd}$ : $p_{unc} < 0.001$, $k$ = FWEc = 692 voxels, $3.40 < t < 10.85$; $2^{nd} > 1^{st}$ : $p_{unc} < 0.001$, $k$ = FWEc = 246 voxels, $3.40 < t < 10.24$). Colourbars indicate t statistic range (red: $1^{st} > 2^{nd}$ video, blue: $1^{st} < 2^{nd}$ video). ACC anterior cingulate cortex, PCC posterior cingulate cortex, M1 primary motor cortex, S1 primary somatosensory cortex, MTG middle temporal gyrus, Ins insula, Str striatum, Tha thalamus, AG angular gyrus, Precu precuneus cortex.

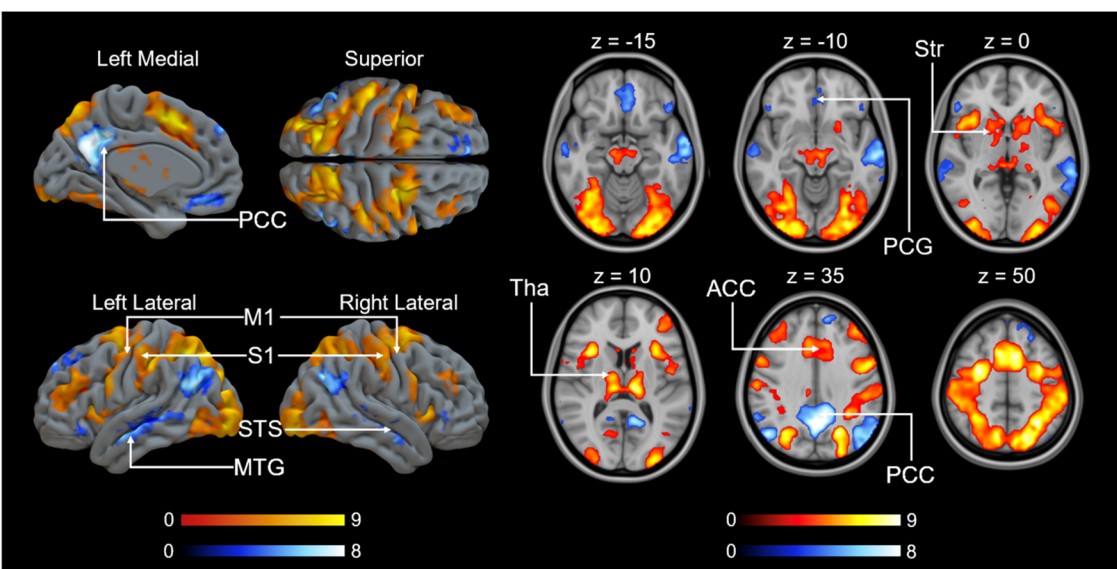

**Fig. 6 | Brain activation for the low-level shock intensity trials in 1st video versus the 2nd video.** The data are thresholded at $p < 0.001$, FWE corrected at the cluster level ($1^{st} > 2^{nd}$ : $p_{unc} < 0.001$, $k$ = FWEc = 146 voxels, $3.40 < t < 9.99$; $2^{nd} > 1^{st}$ : $p_{unc} < 0.001$, $k$ = FWEc = 112 voxels, $3.40 < t < 10.40$). Colourbars indicate t statistic range (red: $1^{st} > 2^{nd}$ video, blue: $1^{st} < 2^{nd}$ video). ACC anterior cingulate cortex, PCC posterior cingulate cortex, M1 primary motor cortex, S1 primary somatosensory cortex, STS superior temporal sulcus, MTG middle temporal gyrus, PCG para-cingulate gyrus, Str striatum, Tha thalamus.

(Supplementary Fig. S4), independently of what donation was given. As in the previous analysis, we used the 1st PC scores for the MOR availability maps to predict haemodynamic activation during the donation phase. Unlike the negative associations between MOR and the parametric modulator for donation size during the 1st video, this analysis revealed positive correlations between BOLD signal and MOR availability particularly in ACC and hippocampus (Fig. 8).

## Discussion
Our main finding was that individual differences in the endogenous opioid system tone do not directly alter participants' decisions to help others, but

are linked with brain activity differences during pain observing and costly, altruistic helping. Our subjects were generally willing to give up significant amounts of money to help others and donated larger sums when they saw the confederate experiencing stronger pain. Trials in which participants donated more money were associated with increased BOLD activity in regions associated with empathy and the central nociceptive system (Ins, ACC, PCC, STS). In line with studies showing that reduced baseline MOR availability is associated with heightened sensitivity to pain[48], baseline MOR availability was negatively correlated with the BOLD responses while witnessing the pain of others in 1st video associated with donating to help. Donation depended on how much pain was displayed in the 1st video, and

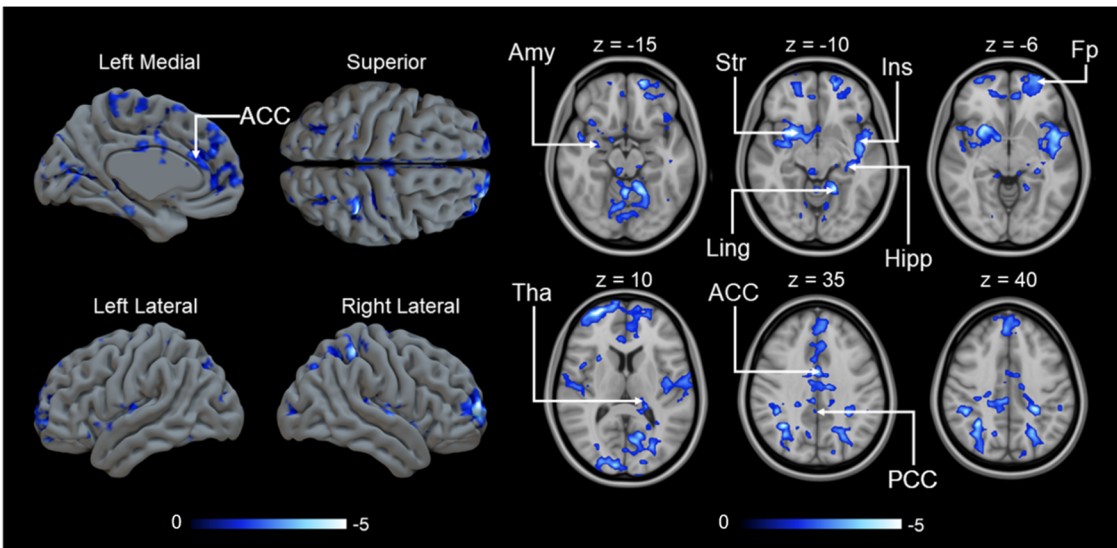

**Fig. 7 | Negative correlation between the first component of MOR availability and the parametric modulator of BOLD response to donation size during 1st video.** ($p < 0.05$, FEW corrected at cluster level, $p_{unc} < 0.05$, $k$ = FWEc = 1800 voxels, 1.78 < $t$ < 9.01). Amy amygdala, Str striatum, Ins insula, Hipp hippocampus, Ling lingual gyrus, Fp frontal pole, Tha thalamus, ACC anterior cingulate cortex, PCC posterior cingulate cortex.

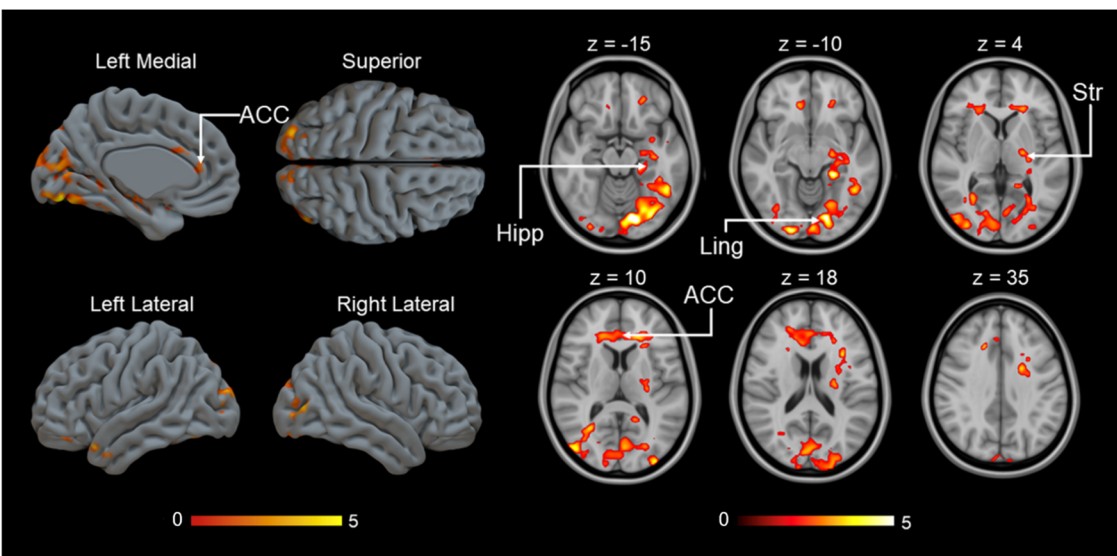

**Fig. 8 | Positive correlation between MOR availability and BOLD response during the donation phase.** ($p < 0.05$, FWE corrected at cluster level, Donation: $p_{unc} < 0.05$, $k$ = FWEc = 10316 voxels, 1.78 < $t$ < 8.44). Hipp hippocampus, ACC anterior cingulate cortex, Ling lingual gyrus, Str striatum.

participants with reduced MOR availability had activity in circuits associated with empathy and nociception that were more strongly associated with donation. Also, brain activity during the donation phase, after they seeing how much pain the other participants expressed, was associated with endogenous MOR tone in hippocampus, ACC, and striatum. These data provide the first direct in vivo evidence for the engagement of the opioid system in the neural processes occurring during costly helping, significantly extending the role of MORs in human social behavior.

As our goal was to model authentic altruistic helping that would also be costly to the participant, we used a naturalistic interactive setting in which we made participants believe that they were interacting with a real person whom they also met before the experiment. Behavioral data confirmed that the manipulation successfully induced costly helping: Participants donated more money when the shock intensity increased and the confederate expressed more intense pain. This finding accords

with previous studies[31] and suggests that people are willing to altruistically help strangers that they have met only recently. FMRI data revealed that witnessing the pain of the others during the 1st video evoked widespread cortical and subcortical activation, in regions associated with empathic pain (aIns, ACC, M1, S1, thalamus, amygdala, and striatum). The results are consistent with meta-analyses of brain regions associated with empathy for pain[19–21]. ACC, PCC, insula, and STS response during 1st video were stronger in trials in which participants later decided to donate more money. Within the limitations of fMRI, activity in these regions may thus have played a role in motivating costly helping. This dovetails with findings from previous studies that showed BOLD activity in similar networks of regions scaled with perceived pain in Hollywood-type videos with various intensities of pain[48], and for the insula, recent intracranial recordings showing that the power in broadband gamma and the spiking of single neurons in this region scales with the perceived pain

intensity for similar stimuli[69]. Importantly, this also matches findings using a similar task acquired in a different lab[52].

We observed a negative association between cerebral MOR availability and the relationship between BOLD activity and donation while viewing 1st video across a wide range of brain regions involved in witnessing pain in three different ways. This association was significant in regions associated with empathy for pain (insula, ACC, thalamus, amygdala, and striatum). This negative association accords with previous findings associating lower MOR availability with higher sensitivity to pain[47], more distress[41] as well as acute adverse emotions evoked by witnessing the pain of others[48]. Previous studies have indicated that the opioid system plays a role in the processing of facial expressions[72,73]. In our experiment, we utilized facial expressions and body movements as experimental stimuli, and our findings further substantiates previous studies. Taken together these data suggest that lower MOR availability makes individuals more sensitive to suffer of others. These data also accord with the general role of the opioid system in maintaining social bonds and attachment[49,51,74], which was here conceptualized as hemodynamic responses to seeing others in distress.

A peculiarity of the paradigm developed by Gallo et al.[31] employed here is that participants can directly monitor the effectiveness of their costly helping by comparing 1st video and 2nd video. Comparing the brain activity across the 1st and 2nd videos (i.e., before and after helping), we found that the 1st video evoked stronger activation in striatum, thalamus, ACC, M1, and S1, whether we compared all 1st video against all 2nd video, or repeat the analyses selecting instances in which 1st video was of low intensity. This pattern was spatially similar to that elicited by the 1st video only. One interpretation why the activity was lower in 2nd video than 1st video (even if the two videos showed similar levels of pain, Fig. 6) is that the opportunity to help decreased the vicarious pain response. This would accord with the negative state relief model for altruism, which states that helping others makes the helper feel better[10,75]. Altruistic behavior could thus be motivated by the (anticipated) alleviation of vicarious pain. Some have argued that altruism is driven by the rewarding nature of empathy and helping[9,11,30], which might predict helping-induced activations in the brain's striatal reward systems (Fig. 5 and Supplementary Fig. S3), which we, however, failed to find in our contrast between the 1st and 2nd videos.

Accordingly, within the limits of reverse inference, the neural data are perhaps best explained by the notion that helping is motivated by an anticipation of reduced vicarious pain/distress rather than by the anticipation of reward. It should, however, be noted that several alternative explanations could account for the observed reduction in brain responses from 1st video to 2nd video. First, 1st video is relevant to the task given to the participants, and the association between the intensity of pain perceivable in 1st video and the donation confirms that participants adapted their responses to the content of 1st video. In contrast, 2nd video is task-irrelevant. Task relevance and the attention it commands, may thus account for the intensity of the response to 1st video. The occipital activation serves as a compelling illustration. Despite the similarity in content between the low shock intensity in the 1st video and 2nd video, greater activation of the visual cortex is apparent in the former. This observation suggests a heightened demand for cognitive resources during this phase. Second, the intensity of 1st video is unpredictable, while the intensity of 2nd video is predictable based on 1st video and the donation. Given that a network similar to the one we observed here has been shown to encode prediction errors while witnessing the pain of others[32], this difference in predictability may also account for the differences in BOLD responses. Together, the observed differences in BOLD activity across 1st video and second should thus be interpreted with caution.

As mentioned in the introduction, networks supporting costly helping have been described, but the role played by the opioid system remains poorly understood. Here we demonstrate that individual differences in baseline $\mu$-receptor availability also relate to the individual differences in brain activity while participants plan and report their costly-helping decision. We observed a predominantly positive correlation between cerebral MOR availability and an extensive BOLD response in the helping phase (donation) in hippocampus, ACC, and striatum. Prior PET studies have

illustrated the modulatory role of the opioid system in emotion, vicarious pain, positive social interaction, prosocial motivation in humans, and pro-social behavior in monkeys[37,39,40,48]. Our findings thus broaden our knowledge of the function of the opioid system, demonstrating its modulating role in the neural processing of costly helping. Importantly, MOR availability had, in general, opposite relationship with haemodynamic responses in the cortical and subcortical pain and emotion circuits during witnessing pain (negative) and deciding to help (positive). This suggests that when witnessing pain, the MORs may act as a buffer against the stress evoked by seeing others in distress. Yet when relieving others' stress via one's own decisions becomes possible (here during donation phase), individuals with high MOR tone show amplified hemodynamic responses. Thus, individuals with high MOR tone might shift their brain activity from the moment of witnessing other people's sufferance (Video1) to the moment of deciding to help them (decision phase). This suggests that MOR tone could be an important molecular pathway modulating altruism and sociability via multiple mechanisms. To delve deeper into the buffering effect of affective stress, we conducted additional analyses and discovered that MOR availability does indeed play a role in modulating the stress response following the observation of others in pain. For further details, please refer to the supplementary materials (Modulation effect of MOR availability to affective stress, Supplementary Figs. S6 and S7).

Contrary to our findings that individual differences in brain activity during costly helping are associated with differences in MOR availability, we did not find a significant correlation between MOR availability and donation size. This is slightly unexpected given the association at the neural level with brain regions associated with donation size, but in general, brain-behavior relationships require substantially larger samples than what can be achieved with invasive experimental PET studies[76]. It is not uncommon that differences in neural activity or receptor availability exist in the absence of observable behavioral differences. The opioid system may not exert a substantial impact on specific behaviors, but could still play a role in shaping broader behavioral tendencies. Besides, it is possible that some other neuromodulator system (such as oxytocin) would be more critical for modulating this type of behavior. However, given the behavioral pharmacological studies pointing towards causal effects of opiates on helping[45] our data contributes to a nascent literature associating the MOR system with brain process of prosocial decision-making.

## Limitations

In a single PET scan, it is impossible to demonstrate the exact molecule-level mechanism for altered receptor availability[36]. Our single PET scan study design only allowed quantifying differences in baseline receptor binding, but not the capacity for endogenous opioid release. We also only studied females so the results may also not generalize to males, given sex differences in empathy[54,56] and MOR availability[36]. Only 14 individuals participated in the PET scan, which may limit the robustness of the results due to the relatively small sample size. In our experiment, not all participants fully believed our cover story—four participants expressed varying degrees of doubt. This may have introduced confounds to the results. However, since helping behavior is common in humans[77] and the participants did not fully disbelieve the manipulation, it is unlikely that this would be a major confound for the study. Another limitation of our experiment is the absence of a control condition. In future studies, incorporating a control condition would be beneficial to better distinguish the emotional responses associated with observing pain from those related to costly helping. Finally, PET and fMRI data were not measured simultaneously, yet prior studies have established that [11C]carfentanil has excellent test-retest reliability even with multiple-months intervals.

## Conclusion

Placebo analgesia studies have suggested that the opioid system may contribute to costly helping decisions. We provide evidence that individual differences in $\mu$-opioid baseline availability can explain significant individual variability in how the brain processes the distress of others in a costly

helping context, and how it processes the decision to help. Brain regions associated with empathic pain such as the aIns, ACC, thalamus, amygdala, striatum, were significantly activated while perceiving the pain of others, and more so on trials in which participants later decided to help more. Activation of these regions decreased following helping. MOR availability was negatively correlated with the processing of the pain of others but dominantly positively correlated with neural responses while making the decision to help. These results suggest that the opioid system is intimately involved in witnessing pain and neural processing of later helping decisions.

## Data availability
Due to the hospital data confidentiality, the source data is not available. However, the group level results are accessible on NeuroVault (https://identifiers.org/neurovault.collection:14151).

## Code availability
The analysis code is available on GitHub. You can access it here: (https://github.com/ChenUTU/Endogenous-opioid-receptor-system-mediates-costly-altruism-in-the-human-brain).

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

## Acknowledgements

This work was supported by grants from the Academy of Finland (#332225), Sigrid Juselius Stiftelse, Signe och Ane Gyllenberg's Stiftelse, and China Scholarship Council (202106040042), as well as grants from the Dutch Research Council (NWO) to V.G. (VIDI grant 452-14-015) and C.K. (VICI grant 453-15-009).

## Author contributions

L.N., C.K., and V.G. designed the study. K.I. and K.S. set up the experiment. K.S. was the confederate. K.S. and V.P. collected data. J.C. performed data analysis under the supervision of V.P., L.N., C.K., and V.G. J.C. wrote the manuscript under supervision of L.N and V.P. J.H. identified structural brain abnormalities in all participants. All authors gave feedback and approved the final version of the manuscript.

## Competing interests

The authors declare no competing interests.
