## [Transparent Peer Review file · Communications Biology]

Endogenous opioid receptor system mediates costly altruism in the human brain

Corresponding Author: Ms Jinglu Chen

Version 0:

Reviewer comments:

Reviewer #2

(Remarks to the Author)

Chen and colleagues' study employed Positron Emission Tomography (PET) and Functional Magnetic Resonance Imaging (fMRI) in a partially overlapping sample to investigate the relationship between availability of cerebral mu opioid receptors (MOR) and costly altruism and its neural substrate.

I enjoyed reading the manuscript, and I would recommend its consideration for publication in the journal Communications Biology, despite not in the current form. Most of my concerns relate to the lack of clarity regarding methodological details, statistical approaches, and report of the results. I will list them in the following bullet points:

- The paragraph "The current study" at the end of the Introduction section contains no hypotheses at all, but instead summarizes the results and discloses a short interpretation. I would prefer that the Authors would clearly state what were their hypotheses before running the study and save the interpretation of their results for later.
- One crucial issue of many Social Neuroscience studies in the lab is represented by the need to lie to the participants and debrief them once the session is over. Nevertheless, it is extremely important to check that participants believed they were doing what the experiments told them to do. In other words, did the Authors check if the manipulation was successful by explicitly asking the participants if they really believed that the confederate was a real participant receiving the shocks and their actions impacted their state? Relying exclusively on the behavioral results to check this sounds inappropriate. In case this was not done, it should be reported as a limitation of the study.
- The exact details of the pain observation task are a bit hard to understand, and this is a major concern not only for the sake of clarity, but also for reproducibility. First, based on whose pain scale were the intensities of expressed pain presented? Second, from Figure 2 it seems possible for participants to offer more than the needed to cancel the pain, which would add another cognitive component to the task, i.e., uncertainty, perhaps loss aversion. This in turn adds an extra layer of interpretation to all the brain contrasts which include the first video. Furthermore, the Authors write that 2 runs with 15 trials in each were presented, but the list of stimuli in the Supplementary Information contains 50 stimulations (17 of intensity 2, 12 of intensity 3 etc.); what does it mean that this were presented with "fixed proportion", and why this was chosen? Were all participants presented with the same 30 stimuli? Was stimuli presentation order randomized across participants?
- When reporting the behavioral results, why was linear mixed model performed (and not repeated measures)? Also, did you test different models to decide the best fitting one? I find it uninformative to know the variance and the SD of the subject random effect; it is more important to know if this effect was significant or not.
- I could not find tables with MRI coordinates for all the analyses, which are a very important source of information. For instance, looking at Fig.3, it seems that the insula activation is more posterior than the usual empathy/vicarious pain activation (differently from most of the studies cited when the Authors report this results). Related to this, insular activation disappears when looking only at low intensity stimuli in the 1st vs. 2nd video contrast. Do the Authors have explanations about this lack of consistency regarding insular activity in their task?
- Why once calculated the cluster size K , the statistical maps are thresholded at $K-1$? And why do positive and negative activations have different cluster thresholds?
- Please state in figure captions what the bars in fMRI figures represent. Why there is always two of them with the min values 0 and 1, respectively? I am especially confused by this in Figure S2, where the max value is the number of ROIs and is associated with color white. How should this be interpreted?
- I very much agree with the caution on interpreting the 1st vs 2nd video, and I especially like the task relevance theory. In this regard, please note the higher Occipital activation in 1st video, probably indicating higher processing and resource allocation during this phase of the experiment.
- Finally, I would like to invite the Authors to spend more words to speculate about the discrepancy between behavioral and brain results. At the end of the day, if the behavioral output is not impacted by differential brain activity and MOR receptor availability, one could speculate that these are not that crucial for this specific behavior, or perhaps alternative strategies (e.g., demand characteristic) or neurochemical systems play an important role in determining behavior.

Minor suggestions (in bold):

Methods, Participants paragraph:

“... and their altruistic behavior is are not influenced by trait harm sensitivity.”

Methods, MRI data acquisition and analysis paragraph:

- “... by using boxcar regressors convolved with HRF in the design matrix.”
- “Additionally, a subtraction contrast was computed for 1st vs. 2nd video and the parametric effect of donation size for 1st video”. This sentence is just not clear.

Reviewer #3

(Remarks to the Author)

This manuscript provides an important insight how the opioid system modulates the responses to vicarious pain responses and the pro-social act to stop this pain for the other. Combination of PET imaging with BOLD responses reveals a unique data-set that connect individual differences in neuropharmacological pathways with a pro-social paradigm.

Here are my comments

1) What is percentage of participants that did not believe the cover story of the confederate?

2) What is the development of the donation over time? The rewarding outcome to keep 10% of the undonated money may not be processed as a stable outcome across trials. Prospect theory suggests 1) reference dependence and 2) diminishing sensitivity of gaining rewards (Kahnemann & Tversky 1979). Diminishing sensitivity implies a convex diminution of reward sensitivity and hence a decreasing relative value of the rewarding outcome of the 10% of the money. This decreasing relative value, rather than the absolute value, has been found to influence attention (Kim & Beck, Psychonomic Bulletin & Review (2020)). Reward learning paradigms have counteracted this by, for example, only monetarizing a few outcomes that are randomly distributed across trials, and/or pay an extra bonus if the participants stay alert throughout the whole experiment (extra payment is made in Möhring & Gläscher Cell Rep 2023). Have the authors tested 1) if there is a change in pro-social helping across time; 2) how participants process the reward (e.g., state or trait measures, as well as rating of the reward) and 3) if there could be a diminishing reward across time? It is important to disentangle probably competing reward processes of social rewards (helping others) and monetary rewards (getting the undonated money).

4) The comparison between neural activation to the 2nd (after donation) and 1st observation (before donation) of pain revealed decreased activation in the 2nd observation. The 1st observation is parametrically modulated by the donation sum and the authors thereby state in the discussion: “MOR availability had activity in circuits associated with empathy and nociception that were more strongly associated with donation “. In order to support this notion, it would be helpful to examine if the association with the 1st pain without the modulation of the donation sum is not existent.

4) There is an interesting effect consisting of a positive correlation between MOR availability and BOLD responses donation. It would be helpful to enter a parametric modulator here, as well (which might be done in a separate first-level to avoid collinearity with the parametric modulator of the 1st pain). Otherwise, the effect of the correlation might simply be due to the offset of the vicarious pain response, but not related to the act of pro-social helping.

5) The authors state that „This suggests that during vicarious pain perception, the MORs may act as a buffer against the stress evoked by seeing others in distress.” This hypothesis might also include that the responses to 1st Pain are reduced as a function of trials, as vicarious pain would induce actions of endogenous opioids that buffer responses to upcoming vicarious pain (Haaker et al. 2017). The authors could explore this buffering of vicarious pain responses across trials.

6) There is evidence that MORs are related to processing of facial expressions in humans (zhao et al. Psychophysiology 2021; Meier et al. Psychoneuroendocrinology 2016; Løseth et al. Psychoneuroendocrinology 2018; Wardle Soc Neuroscience 2006) . Could it be that an opioid effect on processing facial expression might underlie the modulation of socio-affective processes, such as empathy?

Since the authors mention empathic processing, is this study including empathic ratings of the pain and are these empathy ratings related to MOR availability?

Minor:

“brain activity during the donation phase, after they seeing how much pain the other participants expressed, depended on endogenous MOR tone in hippocampus, ACC, and striatum.” This is a correlation between both measures. „dependet on“ might be replaced with “was associated”

Reviewer #4

(Remarks to the Author)

Review of: Endogenous opioid receptor system mediates costly altruism in the human brain

This study presents an interesting, timely and innovative experiment on (financial) helping behavior and the mu-opioid system. The overall design, a combined PET/fMRI approach, is a nice and suitable approach. Participants watch videos of another person receiving shocks and are asked to donate money to reduce shock-level of the next shock the other person receives. Participants donate more money when the shock they saw was more intense, but this did not relate to MOR availability as assessed with PET. FMRI analyses and PET-informed fMRI analyses are subsequently used to assess the brain's response to seeing pain in others and subsequent donation behavior. The manuscript is well-written, but unfortunately the fMRI design and analysis strategy have some important flaws that make the interpretation in terms of costly helping problematic.

1. The fMRI design seems to lack a control condition, for example a video-clip without shock. Consequently, BOLD responses to the first videos (Figure 3) mostly reflect seeing videos and not seeing someone in pain. A (not ideal) solution could be to re-analyze the data using a parametric modulation of shock intensity in the first videos.
2. Following from the above, and given the relationship between shock-level and donation, Figure 4 might indeed reflect BOLD responses to seeing pain in others. This figure is however interpreted as reflecting what part of processing of the pain in others relates to subsequent donations. If the authors want to assess such an effect it is important to show how donations modulate BOLD responses over and above how seeing other's pain does (e.g. by having a control condition or using the parametric modulation from above as comparison).
3. Similarly, Figure 7 might not reflect how MOR availability relates to donation, but again, how MOR availability relates to seeing pain in others.

Minor issues:

4. Sample size is quite low, especially for the PET analyses, which should be mentioned in the limitation section.
5. Could you explain why there is so much variability in time between the fMRI and PET sessions?
6. Did you assess whether, or to what extent, the participants believed that the confederate was a real participant?

Version 1:

Reviewer comments:

Reviewer #2

(Remarks to the Author)

I would like to thank the Authors for their effort in addressing my concerns. It was a pleasure to revisit the updated manuscript. I think it reads better and has been enriched with important details and analyses. Nonetheless, I think some of my comments still need to be addressed more effectively.

1) I appreciate that hypotheses were added to the Introduction section. Still, the hypotheses regarding the relationship between MOR availability and prosocial responses are very vague. Considering the literature presented in the introduction (e.g., Hartmann et al., 2022), one reasonable hypothesis would have been a negative relationship (contrary to what was found in this study). Did the Authors have specific hypotheses in terms of directionality in this regard? If not, please state it clearly and justify why that was the case.

2) Thanks for reporting the results of the credibility test. I find intriguing that, despite not believing the cover story, four participants showed similar behavior to the rest of participants. Could this be interpreted as an effect of social reputation or compliance? And does this create a problem to the interpretation of the overall results? In other words, how can we be sure to be observing correlates of prosocial behavior and not of other psychological phenomena?

3) The Authors interpret the lack of higher insular activation to low stimulations during 1st video compared to 2nd video as the product of similar engagement for these stimulus intensities, whereas in the global 1st vs 2nd video contrast, higher insular activation is interpreted as a tracking of donation. If this is the right interpretation, what is different about the other major affective empathy area then, namely ACC? ACC it is both tracking donation size during 1st video (Figure 4), but also more responsive to low stimulations during 1st video compared to the second video (Figure 6).

Reviewer #3

(Remarks to the Author)

The authors provided amendments to their initial manuscript that addressed the majority of my comments. Two minor points remain to be addressed in relation to my previous comments:

-The authors found for the donation an interaction between shock-level and trial, which could be stated in the supplement.

-The authors conducted separate analyses for the first and the second sessions of the experiment examining the relationship between MOR availability and BOLD response to the 1st video, utilizing donation size as a regressor. The results for the first session were similar to those reported in the paper, but not for the second session. This would suggest that the release of endogenous opioids upon the initial presentation of vicarious pain serves to buffer responses to later instances of vicarious

pain. This has been found for vicarious pain (Haaker et al. Nature Comm 2017), as well as for direct pain (Eippert et al. Journal of Neurosc 2013) and aligns with established theories of endogenous opioids for learning from aversive outcomes (McNally et al. Trends Neurosc. 2011). It would be beneficial to include a mention of the additional analysis conducted for the first and second sessions in the main text of the manuscript, as this is now only available in the supplement.

Reviewer #4

(Remarks to the Author)

This revised version of the manuscript 'Endogenous opioid receptor system mediates costly altruism in the human brain' has substantially improved compared to the initial submission. The authors replied to my previous comments by analysing BOLD responses to the first video in relation to shock-intensity and essentially show that this provides the same results as analysing BOLD responses to the first video in relation to donation-size. The authors proceed to acknowledge that shock-intensity and donation-size share a substantial amount of variance and that teasing them apart is impossible in the current design, at least partly due to the lack of a good control condition/video. With this in mind, there remain two major issues with regards to the interpretation of the results:

1. Interpretations of the link between BOLD-signal and seeing pain in others should be based on the models with shock-intensity/donation-size as regressor, not on models of video-1 vs baseline.
2. None of the fMRI and PET results should be directly linked to donation-size and, by extension, costly helping/altruism.

Regarding point 1, this means that all statements on brain activity in relation to seeing pain in others should be revised (e.g. in the abstract '... brain activity increased in amygdala and anterior insula, anterior cingulate cortex (ACC), striatum, primary motor cortex, primary somatosensory cortex and thalamus when witnessing others' pain' should read something like '... brain activity increased in anterior insula, anterior cingulate cortex (ACC), precuneus and thalamus when witnessing others' pain'.

Regarding point 2, this means that all statements on links between fMRI (and PET) results and costly helping/altruism should be treated with much more caution. This starts with the title: 'Endogenous opioid receptor system mediates costly altruism in the human brain'. In this study there is no evidence for mediation of MOR availability on costly altruism. True, MOR availability is associated with reductions in the relation between donation-size and BOLD response, but this is intertwined with vicarious pain experience. True, MOR availability is related to activation during the donation-decision, but this is unrelated to the actual decision made. So, evidence that MOR availability mediates costly altruism is highly circumstantial, and statements like that should be toned down strongly (see also the final sentences of the abstract and discussion).

Regarding both points it is important to mention the lack of a control condition and the intertwinement of pain processing and donation-size in the limitation section, as well as that a specific design that can assess costly helping independently from the emotional response to seeing pain in others is needed to investigate the true relationship between neural activity and costly helping.

Version 2:

Reviewer comments:

Reviewer #2

(Remarks to the Author)

Thank you for addressing my last comments, I am totally satisfied with your answers. I strongly recommend for this interesting study to be published in Communications Biology.

Reviewer #4

(Remarks to the Author)

Of the two comments that remained after the previous revision the authors answered the second one satisfactorily. To the first comment the authors reply that they would like to leave the interpretation to the reader, partly because of the added complexity due to the new parametric analyses. Although I agree with the complexity standpoint, I remain skeptical about the main analyses and think that presenting them in the abstract without nuance can result in over-interpretation by the readers. Particularly, readers might think that it is a modulation of observed pain that increases amygdala and striatum activity, although there is no evidence for that claim. A simple edit to the text might help in this respect though, as the wording 'brain activity increased in...' implies an association with level of pain. When rephrasing this to 'brain activity was observed in...' the connotation of a pain modulation is reduced, and the wording is more in line with what is actually observed in this analysis.

Thus, please rephrase:

'...brain activity increased in amygdala and anterior insula, anterior cingulate cortex (ACC), striatum, primary motor cortex, primary somatosensory cortex and thalamus when witnessing others' pain.'

To:

'... brain activity was observed in amygdala and anterior insula, anterior cingulate cortex (ACC), striatum, primary motor cortex, primary somatosensory cortex and thalamus when witnessing others' pain.'

Thank you for all the questions. We have answered each question in red text, and the parts we changed in the manuscript are underlined, with the respective pages marked.

Reply to Reviewer #2

- The paragraph "The current study" at the end of the Introduction section contains no hypotheses at all, but instead summarizes the results and discloses a short interpretation. I would prefer that the Authors would clearly state what were their hypotheses before running the study and save the interpretation of their results for later.

Reply: Thank you for the suggestion. We have now provided the hypothesis at the Introduction:

P5: The current study

We expected that participants would show a propensity to donate money to alleviate the pain of the confederate, as suggested by prior research (Ioumpa et al., 2024). Additionally, previous studies suggest a potential negative correlation between MOR availability and brain regions associated with empathy and pain (Karjalainen et al., 2017). Therefore, we anticipate observing a consistent negative correlation between MOR availability and viewing others' pain. Last, we hypothesized that variations in MOR availability may influence altruistic tendencies among individuals at both behavioral and neural levels.

- One crucial issue of many Social Neuroscience studies in the lab is represented by the need to lie to the participants and debrief them once the session is over. Nevertheless, it is extremely important to check that participants believed they were doing what the experiments told them to do. In other words, did the Authors check if the manipulation was successful by explicitly asking the participants if they really believed that the confederate was a real participant receiving the shocks and their actions impacted their state? Relying exclusively on the behavioral results to check this sounds inappropriate. In case this was not done, it should be reported as a limitation of the study.

Reply:

Consistent with previous research, we asked participants at the end of the experiment, 'Do you think the experimental setup was realistic enough to believe it?' on a scale from 1 (strongly disagree) to 7 (strongly agree) (Gallo et al., 2018). Most participants felt it was realistic and provided ratings over 4 (Fig. 1).

Figure 1. Experiment credibility test. The horizontal axis represents the degree of trust, the vertical axis is the number of participants. The number above the bars is the proportion of participants at that value to all participants.

Four participants questioned the authenticity of the experiment to some extent. To investigate whether their behavioral patterns differed from those of others, we conducted a GLM analysis for these four participants only. The results remained significant, $\beta = 0.827$, $SE = 0.073$, $t = 11.285$, $p < 0.001$ (Fig. 2a).

Thus, even though these participants did not fully believe the cover story, they still carried out costly helping behaviour similarly as others (See Fig. 2b, which is the Fig. 2 in the manuscript).

Figure 2a. Donation size increased with increasing shock intensity as other participants. Each dot represents the average donation of a participant for all the stimuli of that intensity. $N=4$ participants. The box represents the 1st, 2nd and 3rd quartile, the whiskers the highest and lowest value within 1.5 interquartile range.

Figure 2b. Donation size increased with increasing shock intensity (Figure 2 in the manuscript).

- The exact details of the pain observation task are a bit hard to understand, and this is a major concern not only for the sake of clarity, but also for reproducibility. First, based on whose pain scale were the intensities of expressed pain presented? Second, from Figure 2 it seems possible for participants to offer more than the needed to cancel the pain, which would add another cognitive component to the task, i.e., uncertainty, perhaps loss aversion. This in turn adds an extra layer of interpretation to all the brain contrasts which include the first video. Furthermore, the Authors write that 2 runs with 15 trials in each were presented, but the list of stimuli in the Supplementary Information contains 50 stimulations (17 of intensity 2, 12 of intensity 3 etc.); what does it mean that this were presented with “fixed proportion”, and why this was chosen? Were all participants presented with the same 30 stimuli? Was stimuli presentation order randomized across participants?

Reply:

The stimuli were created similarly as in the previous study on costly helping (Gallo et al., 2018). To maintain consistency across experiments, specific instructions were provided to the confederate on how to express pain at different levels. Following the recording of all videos, their quality was assessed and compared to the videos in Gallo's paper. As a result, the pain scale used here matches that of Gallo's paper and was based on the observers' evaluation of the pain of the confederate in the video.

While we acknowledge that the experience of vicarious pain is not solely attributable to the observation of the first video, it serves as a primary component of the observed response. This is why we utilize this component to represent brain activation. However, there are additional cognitive elements at play simultaneously. Consequently, we have revised the paper accordingly:

P12: Results

Subtitle changes from “BOLD-fMRI responses to vicarious pain perception” to “BOLD-fMRI responses to confederate’s initial pain”

Figure 3. title: Main effect of brain responses to vicarious pain during the first video.

P17: Discussion

Our main finding was that individual differences in the endogenous opioid system tone do not directly alter participants’ decisions to help others, but are linked with brain activity differences during pain **observing** and costly, altruistic helping.

P17: Discussion

Subtitle changes from “Neural basis of vicarious pain perception” to “Neural basis of witnessing the pain of others”

P18: Discussion

Subtitle changes from “Endogenous opioids and vicarious pain perception” to “Endogenous opioids and witnessing pain”

We observed a negative association between cerebral MOR availability and the relationship between BOLD activity and donation while viewing 1st video across many brain regions involved in **witnessing** pain in three different ways.

P19: Discussion: Opioid system modulates brain activity during costly helping

Importantly, MOR availability had, in general, opposite relationship with haemodynamic responses in the cortical and subcortical pain and emotion circuits during **witnessing** pain (negative) and deciding to help (positive). This suggests that when **witnessing** pain, the MORs may act as a buffer against the stress evoked by seeing others in distress.

P20: Conclusion

These results suggest that the opioid system is intimately involved in **witnessing** pain and neural processing of **later** helping decisions.

P29: Supplementary Materials

Subtitle changes from “Vicarious pain responses to 2nd video” to “Witnessing pain after costly helping”

Finally, regarding the “fixed proportion” in the experiment, it is important to note that the distribution of pain levels was consistent for all participants in the initial video they viewed, with an even balance across the various levels. This balanced ratio was maintained across multiple shock intensities, as our focus is on understanding the relationship between shock intensity and donation, as well as quantifying how the brain processes varying levels of pain induced by shock intensity. This necessitated the inclusion of multiple shock intensities. The video pool consisted of 17 videos at intensity 2, 12 at intensity 3, 9 at intensity 4, 7 at intensity 5, and 5 at intensity 6. These videos were initially presented to participants in a random order without repetition. Following the initial presentation of all videos, if videos corresponding to specific intensity levels needed to be presented again, they were shown repeatedly.

- When reporting the behavioral results, why was linear mixed model performed (and not repeated measures)? Also, did you test different models to decide the best fitting one? I find it uninformative to know the variance and the SD of the subject random effect; it is more important to know if this effect was significant or not.

Reply:

In our view there is no obvious reason for choosing repeated measures ANOVA over LMM given the latter’s superior flexibility, ability to handle missing and unbalanced data, and the capacity to model individual variability more accurately. Furthermore,

repeated measures ANOVA requires calculating the mean of the dependent variable for each level of the independent variable, which is not only slightly cumbersome, but can obscure individual variations. Finally, the requirement of sphericity in ANOVA, which LMM does not have, poses additional constraints and can inflate Type I errors if violated. Although none of the above is a major issue here, where we simply analyzed the effect of pain intensity on donation size, we feel LMM is preferable over repeated measure ANOVA. In any case, repeated measures ANOVA also produces a highly significant main effect of pain intensity on donation size ($F(4, 116) = 123.042, p < .001$).

It is not straightforward to calculate a p-value for the random effect and instead it is more informative to report the variance/standard deviation associated with random effects than to test their significance using p-values. Therefore, reporting the SD of the random effect is customary as it directly quantifies the between-subject variability, which is often of primary interest in mixed-effects modeling.

Regarding model comparison, we compared the cases where only the intercept is a random variable (Model 1) and where both the intercept and slope are random variables (Model2). The results show that the model where both are random variables performs better, $p < 0.001$ (Table 1). Therefore, we will replace the models in the paper accordingly.

	AIC	BIC
Model 1	2746.1	2765.2
Model 2	2607.3	2636.1

Table 1. Model comparison results. Model 1 sets the intercept as a random variable, Model 2 sets both the intercept and slope as random variables. Smaller AIC and BIC represents better fit to the data.

- I could not find tables with MRI coordinates for all the analyses, which are a very important source of information. For instance, looking at Fig.3, it seems that the insula activation is more posterior than the usual empathy/vicarious pain activation (differently from most of the studies cited when the Authors report this results). Related to this, insular activation disappears when looking only at low intensity stimuli in the 1st vs. 2nd video contrast. Do the Authors have explanations about this lack of consistency regarding insular activity in their task?

Reply:

we believe that fMRI activation coordinates provide a misleading outlook of the results, as both large and small clusters yield only a single coordinate whose center of mass may not be indicative of the true extent of the cluster. Therefore, we have uploaded all our result files to NeuroVault as nifti files to allow readers to examine them thoroughly in 3D space and incorporate them into meta-analyses.

Figure 3 illustrates our comparison between the detected activations and the anatomical location of the insula, as well as against findings from prior studies (Caspar et al., 2022; Zhou et al., 2020). Notably, our results encompass a substantial portion of the insula. In the paper, we used affective vicarious pain signature to explore the relationship between individual MOR differences and responses in the

vicarious pain observation network. We compared our results with the signature, and they also exhibit remarkable similarity with previous investigations (Zhou et al., 2020). Our observed heightened activation may stem from the utilization of a video stimulus in our experiment, whereas preceding studies typically employed static images.

Figure 3. Insula is shown in green in all the images. In the left and middle images, red colour represents the corrected T map for positive effects during the viewing of the 1st video. In the rightmost image, red represents the beta map corresponding to the 1st video viewing, whereas blue represents the beta map from the signature analysis, both focusing solely on positive effects.

Regarding the missing insula activation, figure 4 (in the paper) reveals a higher activation of the insula in response to larger donations, suggesting its sensitivity to donation size. Therefore, it is plausible that insula activation remains consistent between the low-shock intensity in 1st video and 2nd video, given that the latter predominantly consists of low shock intensity content.

We also discussed this in the paper in result part:

P15: Results: BOLD-fMRI response changes following costly helping

One intriguing finding is the apparent loss of activation in the insula. As illustrated in Fig. 4, there is heightened activation of the insula in response to larger donations, indicating its sensitivity to donation size. Therefore, it is conceivable that insula activation remains consistent between the low-shock intensity in the first video and the second video, given that the latter primarily features content with low shock intensity.

- Why once calculated the cluster size K , the statistical maps are thresholded at $K-1$? And why do positive and negative activations have different cluster thresholds?

Reply:

Upon revisiting our code, we realized there was an error. This should read 'k' rather than 'k-1'.

- Please state in figure captions what the bars in fMRI figures represent. Why there is always two of them with the min values 0 and 1, respectively? I am especially confused by this in Figure S2, where the max value is the number of ROIs and is associated with color white. How should this be interpreted?

Reply:

We employed SPM to generate all our results, setting thresholds and applying FWE cluster-level corrections. Subsequently, we used Surfice and MRICroGL software to create surface and layered brain images, utilizing the exact same files for figure generation. However, in Surfice, setting the minimum value to 0 results in the entire brain being rendered in color. To avoid this, we set the minimum value to 1. Given that all data were FWE corrected, the minimum t value must exceed 1 in this scenario. Importantly, adjusting the minimum value has no impact on the image results; no non-significant effect was included in the plot and no significant effects were left out. To ensure clarity for our readers and avoid any potential misinterpretation, we modified the minimum value to 0.

Figure S2 pertains to number of ROIs, so the max value means that all the ROIs have an association.

- I very much agree with the caution on interpreting the 1st vs 2nd video, and I especially like the task relevance theory. In this regard, please note the higher Occipital activation in 1st video, probably indicating higher processing and resource allocation during this phase of the experiment.

Reply: We appreciate your suggestion. Indeed, there are multiple theories that can partially explain this phenomenon. We have now discussed this in the MS as follows:

P19: Discussion: Brain basis of costly helping

First, 1st video is relevant to the task given to the participants, and the association between the intensity of pain perceivable in 1st video and the donation confirms that participants adapted their responses to the content of 1st video. In contrast, 2nd video is task-irrelevant. Task relevance and the attention it commands, may thus account for the intensity of the response to 1st video. The occipital activation serves as a compelling illustration. Despite the similarity in content between the low shock intensity in the 1st video and 2nd video, greater activation of the visual cortex is apparent in the former. This observation suggests a heightened demand for cognitive resources during this phase.

- Finally, I would like to invite the Authors to spend more words to speculate about the discrepancy between behavioral and brain results. At the end of the day, if the behavioral output is not impacted by differential brain activity and MOR receptor availability, one could speculate that these are not that crucial for this specific behavior, or perhaps alternative strategies (e.g., demand characteristic) or neurochemical systems play an important role in determining behavior.

Reply: We have now discussed this in the MS as follows:

P20: Discussion: Opioid system modulates brain activity during costly helping

This is slightly unexpected given the association at the neural level with brain regions associated with donation size, but in general, brain-behaviour relationships require substantially larger samples than what can be achieved with invasive experimental PET studies (Marek et al., 2022). It is not uncommon that differences in neural activity or receptor availability exist in the absence of observable behavioral differences. The opioid system may not exert a substantial impact on specific behaviours, but could still play a role in shaping broader behavioral tendencies. Besides, it is possible that some other neuromodulator system

(such as oxytocin) would be more critical for modulating this type of behaviour.

Minor suggestions (in bold):

- Methods, Participants paragraph:
"... and their altruistic behavior is are not influenced by trait harm sensitivity."

Reply: Noted and changed

- Methods, MRI data acquisition and analysis paragraph:
"... by using boxcar regressors convolved with HRF in the design matrix."

Reply: Noted and changed

- Methods, MRI data acquisition and analysis paragraph: "Additionally, a subtraction contrast was computed for 1st vs. 2nd video and the parametric effect of donation size for 1st video". This sentence is just not clear.

Reply: This sentence has been modified as follows
Additionally, a subtraction contrast was computed for 1st vs. 2nd video

Reply to Reviewer #3

- What is percentage of participants that did not believe the cover story of the confederate?

Reply: Please see our reply to Reviewer #1 above.

- What is the development of the donation over time? The rewarding outcome to keep 10% of the undonated money may not be processed as a stable outcome across trials. Prospect theory suggests 1) reference dependence and 2) diminishing sensitivity of gaining rewards (Kahnemann & Tversky 1979). Diminishing sensitivity implies a convex diminution of reward sensitivity and hence a decreasing relative value of the rewarding outcome of the 10% of the money. This decreasing relative value, rather than the absolute value, has been found to influence attention (Kim & Beck, *Psychonomic Bulletin & Review* (2020)). Reward learning paradigms have counteracted this by, for example, only monetarizing a few outcomes that are randomly distributed across trials, and/or pay an extra bonus if the participants stay alert throughout the whole experiment (extra payment is made in Möhring & Gläscher *Cell Rep* 2023). Have the authors tested 1) if there is a change in pro-social helping across time; 2) how participants process the reward (e.g., state or trait measures, as well as rating of the reward) and 3) if there could be a diminishing reward across time? It is important to disentangle probably competing reward processes of social rewards (helping others) and monetary rewards (getting the undonated money).

Reply: Upon incorporating time into our linear mixed model, we observed the following effects:

The main effect of the 'trial' variable was found to be significant, $\beta = -0.026$, $SE = 0.009$, $t = -2.920$, $p = 0.004$).

Additionally, we identified a significant interaction between 'trial' and shock intensity, $\beta = 0.006$, $SE = 0.003$, $t = 2.095$, $p = 0.036$).

Furthermore, our analysis revealed that individuals tend to increase their donation behavior over time when watching videos of shock intensity 6 compared to shock intensity 2, $\beta = 0.026$, $SE = 0.011$, $t = 2.305$, $p = 0.021$) (see Fig. 4).

It's worth noting that although there is a discernible change in subjects' donation behavior at shock intensity 6, this change is relatively small in practical terms and is unlikely to significantly impact the overall results.

Figure 4. The plot illustrates how donation sizes change over time, with different colors representing varying shock intensities. Each line represents a regression line fitted to the data, each dot represents each trial.

- The comparison between neural activation to the 2nd (after donation) and 1st observation (before donation) of pain revealed decreased activation in the 2nd observation. The 1st observation is parametrically modulated by the donation sum and the authors thereby state in the discussion: "MOR availability had activity in circuits associated with empathy and nociception that were more strongly associated with donation ". In order to support this notion, it would be helpful to examine of the association with the 1st pain without the modulation of the donation sum is not existent.

Reply: The suggested analysis revealed that only in the visual cortex there was a discernible relationship between MOR availability and brain activation during the viewing of the first video, thus we have retained the original conclusions.

Figure 5. Positive correlation between the first component of MOR availability and BOLD response to the 1st video.

- There is an interesting effect consisting of a positive correlation between MOR availability and BOLD responses donation. It would be helpful to enter a parametric modulator here, as well (which might be done in a separate first-level to avoid collinearity with the parametric modulator of the 1st pain). Otherwise, the effect of the correlation might simply be due to the offset of the vicarious pain response, but not related to the act of pro-social helping.

Reply: Following the recommendation and utilizing donation size as the modulator in the first-level model for donation phase, the analysis (Figure 6) indicated that the main result remained unchanged. However, it appears that the striatal effects might be modulated by the first video.

Figure 6. Correlation between the first component of MOR availability and the parametric modulator of BOLD response to donation size during donation. Ling = lingual gyrus, Hipp = hippocampus, ACC = anterior cingulate cortex, PCC = posterior cingulate cortex, Ins = insula.

- The authors state that „This suggests that during vicarious pain perception, the MORs may act as a buffer against the stress evoked by seeing others in distress.“ This hypothesis might also include that the responses to 1st Pain are reduced as a function of trials, as vicarious pain would induce actions of endogenous opioids that buffer responses to upcoming vicarious pain (Haaker et al. 2017). The authors could explore this buffering of vicarious pain responses across trials.

Reply: To further explore whether MOR availability modulates affective stress when witnessing others in pain, we computed correlation between the contrast of first session 1st video minus second session 1st video and MOR availability and found positive correlation in caudate and ACC (Fig. 7). We also conducted separate analyses for the first and the second sessions of the experiment examining the relationship between MOR availability and BOLD response to the 1st video, utilizing donation size as a regressor. Interestingly, we observed similar results to those reported in the paper for the first session (Fig. 8). However, in the second session, no significant results were found. This once again suggests that MOR availability may play a role in reducing stress during the experiment.

Figure 7. Positive correlation between the first component of MOR availability and the parametric modulator of BOLD response to donation size during the first session of 1st video subtracted from the second session. ACC = anterior cingulate cortex.

Figure 8. Negative correlation between the first component of MOR availability and the parametric modulator of BOLD response to donation size during 1st video in the first fMRI session. Ling = lingual gyrus, Hipp = hippocampus, ACC = anterior cingulate cortex, PCC = posterior cingulate cortex, Ins = insula, Fp = frontal pole, Tha = thalamus.

We also discussed this in the MS as follows:

P20: Discussion: Opioid system modulates brain activity during costly helping
To delve deeper into the buffering effect of affective stress, we conducted additional analyses and discovered that MOR availability does indeed play a role in modulating the stress response following the observation of others in pain. For further details, please refer to the supplementary materials (Modulation effect of MOR availability to affective stress).

P32: Supplementary Materials: Modulation effect of MOR availability to affective stress

To further explore whether MOR availability modulates affective stress when witnessing others in pain, we computed correlation between the contrast (1st video

in the first session minus 1st video in the second session) and MOR availability and found positive correlation in caudate and ACC (Fig. S6). We also conducted separate analyses examining the relationship between MOR availability and BOLD response to the 1st video, utilizing donation size as a regressor. Interestingly, we observed similar results to those reported in the paper (Fig. 7) for the first session (Fig. S7). However, in the second session, no significant results were found. This once again suggests that MOR availability may play a role in reducing stress during the experiment.

Figure S6. Positive correlation between the first component of MOR availability and the parametric modulator of BOLD response to donation size during the first session of 1st video subtracted from the second session. ACC = anterior cingulate cortex.

Figure S7. Negative correlation between the first component of MOR availability and the parametric modulator of BOLD response to donation size during 1st video in the first fMRI session. Ling = lingual gyrus, Hipp = hippocampus, ACC = anterior cingulate cortex, PCC = posterior cingulate cortex, Ins = insula, Fp = frontal pole, Tha = thalamus.

- There is evidence that MORs are related to processing of facial expressions in humans (zhao et al. Psychophysiology 2021; Meier et al. Psychoneuroendocrinology 2016; Løseth et al. Psychoneuroendocrinology 2018; Wardle Soc Neuroscience 2006). Could it be that an opioid effect on processing facial expression might underlie the modulation of socio-affective processes, such as empathy? Since the authors mention empathic processing, is this study including empathic ratings of the pain and are these empathy ratings related to MOR availability?

Reply: We have now discussed this in the paper as follows:

P18: Discussion: Endogenous opioids and witnessing pain

This negative association accords with previous findings associating lower MOR availability with higher sensitivity to pain (Hagelberg et al., 2012), more distress (Nummenmaa et al., 2020) as well as acute adverse emotions evoked by witnessing the pain of others (Karjalainen et al., 2017). Previous studies have indicated that the opioid system plays a role in the processing of facial expressions (Løseth et al., 2018; Meier et al., 2016). In our experiment, we utilized facial expressions and body movements as experimental stimuli, and our findings further substantiates the previous findings

Minor:

“brain activity during the donation phase, after they seeing how much pain the other participants expressed, depended on endogenous MOR tone in hippocampus, ACC, and striatum.” This is a correlation between both measures. „dependet on“ might be replaced with “was associated”

Reply: Noted and changed.

Reply to Reviewer #4

- The fMRI design seems to lack a control condition, for example a video-clip without shock. Consequently, BOLD responses to the first videos (Figure 3) mostly reflect seeing videos and not seeing someone in pain. A (not ideal) solution could be to re-analyze the data using a parametric modulation of shock intensity in the first videos.

Reply: We employed shock intensity as a regressor to predict brain activity of viewing 1st video (Fig. 9). The insula, thalamus, anterior cingulate cortex, and precuneus cortex—components of the empathy network—are more activated as shock intensity increases when observing others in distress.

Figure 9. Brain regions with BOLD signals associated with shock intensity during the 1st video. Hipp = hippocampus, ACC = anterior cingulate cortex, Ins = insula, Tha = thalamus, Precu = precuneus cortex.

We discussed this as follows:

P12: BOLD-fMRI responses to confederate's initial pain

To discern whether the observed effect stems from witnessing others in pain rather than viewing videos, we utilized shock intensity as a regressor instead of donation size (Figure S8). Further details can be found in the supplementary materials.

P33: Supplementary Materials: Witnessing pain effect from 1st video

We employed shock intensity as a regressor to predict brain activity of viewing 1st video (Fig. S8). The insula, thalamus, anterior cingulate cortex, and precuneus cortex—components of the empathy network—are more activated as shock intensity increases when observing others in distress. This finding suggests that the paradigm elicits emotional changes beyond video viewing.

Figure S8. Brain regions with BOLD signals associated with shock intensity during the 1st video. Hipp = hippocampus, ACC = anterior cingulate cortex, Ins = insula, Tha = thalamus, Precu = precuneus cortex.

- Following from the above, and given the relationship between shock-level and donation, Figure 4 might indeed reflect BOLD responses to seeing pain in others. This figure is however interpreted as reflecting what part of processing of the pain in others relates to subsequent donations. If the authors want to assess such an effect it is important to show how donations modulate BOLD responses over and above how seeing other's pain does (e.g. by having a control condition or using the parametric modulation from above as comparison).

Reply: We agree that donation size is not the ideal parameter for measuring the impact of seeing others in pain. Considering the inherent features of the paradigm, shock intensity and donation size are highly correlated, making it impractical to include both as regressors in the model. Shock intensity is predetermined before the experiment, while donation size reflects the participants' reactions and feelings. Therefore, we believe donation size is a better indicator of participants' feelings than shock intensity. However, we would like to discuss this point further in the supplementary section following the last question:

P33: Supplementary Materials: Witnessing pain effect from 1st video

We used donation size as a regressor to explore the BOLD responses of seeing others in pain rather than shock intensity in the main text. Considering the inherent features of the paradigm, shock intensity and donation size are highly correlated, making it impractical to include both as regressors in the model. Shock intensity is predetermined before the experiment, while donation size reflects the participants' reactions and feelings. Therefore, we believe donation size is a better indicator of participants' feelings than shock intensity. As expected, the main results remain consistent between these two models.

- Similarly, Figure 7 might not reflect how MOR availability relates to donation, but again, how MOR availability relates to seeing pain in others.

Reply: Figure 7 shows the correlation between MOR availability and viewing pain, which is modeled as the function of donation size. Figure 8 is the correlation between MOR availability and donation. Unfortunately, it would be technically

impossible to set up a model that would be devoid of any sort of pain perception at all thus we argue that our design is as well optimized in this regard as possible.

Minor issues:

- Sample size is quite low, especially for the PET analyses, which should be mentioned in the limitation section.

Reply: Noted and added to the Limitations section

P21: Limitations

Only 14 individuals participated in the PET scan, which may limit the robustness of the results due to the relatively small sample size.

- Could you explain why there is so much variability in time between the fMRI and PET sessions?

Reply: This issue primarily pertains to scheduling. Carfentanil exhibits high test-retest reliability (Hirvonen et al., 2009), reducing the likelihood of significant up or downregulation between scans. Even if such changes were to occur, they might only increase the noise in the signal and observed effects, rather than leading to false positives.

- Did you assess whether, or to what extent, the participants believed that the confederate was a real participant?

Reply: Please see our reply to Reviewer 1.

References

- Caspar, E. A., Ioumpa, K., Arnaldo, I., Di Angelis, L., Gazzola, V., & Keysers, C. (2022). Commanding or Being a Simple Intermediary: How Does It Affect Moral Behavior and Related Brain Mechanisms? *ENeuro*, 9(5), 1–24. <https://doi.org/10.1523/ENEURO.0508-21.2022>
- Gallo, S., Paracampo, R., Müller-Pinzler, L., Severo, M. C., Blömer, L., Fernandes-Henriques, C., Henschel, A., Lammes, B. K., Maskaljunas, T., Suttrup, J., Avenanti, A., Keysers, C., & Gazzola, V. (2018). The causal role of the somatosensory cortex in prosocial behaviour. *ELife*, 7, 1–31. <https://doi.org/10.7554/eLife.32740>
- Hirvonen, J., Aalto, S., Hagelberg, N., Maksimow, A., Ingman, K., Oikonen, V., Virkkala, J., Någren, K., & Scheinin, H. (2009). Measurement of central μ -opioid receptor binding in vivo with PET and [11C]carfentanil: A test-retest study in healthy subjects. *European Journal of Nuclear Medicine and Molecular Imaging*, 36(2), 275–286. <https://doi.org/10.1007/s00259-008-0935-6>
- Zhou, F., Li, J., Zhao, W., Xu, L., Zheng, X., Fu, M., Yao, S., Kendrick, K. M., Wager, T. D., & Becker, B. (2020). Empathic pain evoked by sensory and emotional-communicative cues share common and process-specific neural representations. *ELife*, 9, 1–27. <https://doi.org/10.7554/ELIFE.56929>

Reviewers' comments:

Reviewer #2

1) I appreciate that hypotheses were added to the Introduction section. Still, the hypotheses regarding the relationship between MOR availability and prosocial responses are very vague. Considering the literature presented in the introduction (e.g., Hartmann et al., 2022), one reasonable hypothesis would have been a negative relationship (contrary to what was found in this study). Did the Authors had specific hypotheses in terms of directionality in this regard? If not, please state it clearly and justify why that was the case.

Reply: Hartmann's paper illustrates that placebo analgesia reduces costly prosocial helping. However, it does not suggest that between-subject variation in MOR availability directly influence helping behavior. In our study, we used an subject-level index of MOR availability to predict helping. Previous research has shown that MOR availability is negatively correlated with haemodynamic responses to witnessing others in pain and positively correlated with prosocial motivation. We have now amended these hypotheses to the MS:

P4: The current study

Additionally, previous studies suggest a potential negative correlation between MOR availability and brain regions associated with empathy and pain¹. Therefore, we hypothesized a negative correlation between MOR availability and viewing others' pain. Finally, we hypothesized that individuals with high MOR availability will demonstrate greater altruistic tendencies at both behavioral and neural levels, given the positive correlation between MOR availability and prosocial motivation^{50,51}.

2) Thanks for reporting the results of the credibility test. I find intriguing that, despite not believing the cover story, four participants showed similar behavior to the rest of participants. Could this be interpreted as an effect of social reputation or compliance? And does this create a problem to the interpretation of the overall results? In other words, how can we be sure to be observing correlates of prosocial behavior and not of other psychological phenomena?

Reply:

Costly helping is common in humans². Even when imagining others in distress, people tend to exhibit a desire to help. Brain activation patterns during imagined and real helping scenarios share many commonalities³. In our study, a few participants expressed *some* skepticism about the reality of the experiment; however, they did not completely disbelieve it thus the inclusion of them in the sample is in our view warranted.

We will incorporate this discussion into the paper:

P20: Limitations

Only 14 individuals participated in the PET scan, which may limit the robustness of the results due to the relatively small sample size. In our experiment, not all participants fully believed our cover story—four participants expressed varying degrees of doubt. This may have introduced confounds to the results. However, since helping behavior common in humans² and the participants did not fully disbelieve the manipulation, it is unlikely that this would be a major confound for the study.

3)The Authors interpret the lack of higher insular activation to low stimulations during 1st video compared to 2nd video as the product of similar engagement for these stimulus intensities, whereas in the global 1st vs 2nd video contrast, higher insular activation is interpreted as a tracking of donation. If this is the right interpretation, what is different about the other major affective empathy area then, namely ACC? ACC it is both tracking donation size during 1st video (Figure 4), but also more responsive to low stimulations during 1st video compared to the second video (Figure 6).

Reply: Our results indicate that the insular and ACC exhibit sensitivity to donation size. The ACC shows greater activation during the 1st video at low shock intensity, whereas the insular does not demonstrate a significant difference in activation between the low-level shock intensities of the first and second videos. These findings suggest that the insular cortex and ACC may respond differently depending on the level of donation. Furthermore, the ACC appears to be differentially involved depending on whether participants viewed the video before or after donating. However, it is important to note that we cannot conclusively determine that the insular cortex's function is solely limited to tracking donation behavior.

Reviewer #3:

-The authors found for the donation an interaction between shock-level and trial, which could be stated in the supplement.

Reply: We have added this section to the supplement:

P9: Behavioral results

For the intercept, variance is 1.943 and SD 1.394. For the slope, variance is 0.104 and SD is 0.323. Given that the sensitivity of reward from helping behavior may vary over time, we included time as a factor in our model. The results can be found in the supplementary material.

Supplementary:

Temporal influence on donation behavior

Upon incorporating time into our linear mixed model, we observed the following effects:

The main effect of the 'trial' variable was found to be significant, $\beta = -0.026$, SE = 0.009, $t = -2.920$, $p = 0.004$). Additionally, we identified a significant interaction between 'trial' and shock intensity, $\beta = 0.006$, SE = 0.003, $t = 2.095$, $p = 0.036$). Furthermore, our analysis revealed that individuals tend to increase their donation behavior over time when watching videos of shock intensity 6 compared to shock intensity 2, $\beta = 0.026$, SE = 0.011, $t = 2.305$, $p = 0.021$) (see Fig. S9). It's worth noting that although there is a discernible change in subjects' donation behavior at shock intensity 6, this change is relatively small in practical terms and is unlikely to significantly impact the overall results.

Figure S9. The plot illustrates how donation sizes change over time, with different colors representing varying shock intensities. Each line represents a regression line fitted to the data; each dot represents each trial.

-The authors conducted separate analyses for the first and the second sessions of the experiment examining the relationship between MOR availability and BOLD response to the 1st video, utilizing donation size as a regressor. The results for the first session were similar to those reported in the paper, but not for the second session. This would suggest that the release of endogenous opioids upon the initial presentation of vicarious pain serves to buffer responses to later instances of vicarious pain. This has been found for vicarious pain (Haaker et al. Nature Comm 2017), as well as for direct pain (Eippert et al. Journal of Neurosc 2013) and aligns with established theories of endogenous opioids for learning from aversive outcomes (McNally et al. Trends Neurosc. 2011). It would be beneficial to include a mention of the additional analysis conducted for the first and second sessions in the main text of the manuscript, as this is now only available in the supplement.

Reply: Thank you for the suggestion. However, due to the publication policy, which limits the main text to a maximum of 10 figures, we have to include the additional results in the supplementary materials.

Reviewer #4:

1. Interpretations of the link between BOLD-signal and seeing pain in others should be based on the models with shock-intensity/donation-size as regressor, not on models of video-1 vs baseline.
2. None of the fMRI and PET results should be directly linked to donation-size and, by extension, costly helping/altruism.

Regarding point 1, this means that all statements on brain activity in relation to seeing pain in others should be revised (e.g. in the abstract ‘... brain activity increased in amygdala and anterior insula, anterior cingulate cortex (ACC), striatum, primary motor cortex, primary somatosensory cortex and thalamus when witnessing others' pain’ should read something like ‘... brain activity increased in anterior insula, anterior cingulate cortex (ACC), precuneus and thalamus when witnessing others' pain’.

Reply: Thank you for your suggestion. We have completed the analyses based on your previous recommendations. The results can be found in the main text, Figures 3 and 4, as well as in Supplementary Figure S8. We think that the model presented in the main text provides a robust representation of neural activation in response to observing others in pain. While incorporating the regressor for donation size introduces additional complexity, we encourage readers to review all results and figures to draw their own conclusions.

Regarding point 2, this means that all statements on links between fMRI (and PET) results and costly helping/altruism should be treated with much more caution. This starts with the title: 'Endogenous opioid receptor system mediates costly altruism in the human brain'. In this study there is no evidence for mediation of MOR availability on costly altruism. True, MOR availability is associated with reductions in the relation between donation-size and BOLD response, but this is intertwined with vicarious pain experience. True, MOR availability is related to activation during the donation-decision, but this is unrelated to the actual decision made. So, evidence that MOR availability mediates costly altruism is highly circumstantial, and statements like that should be toned down strongly (see also the final sentences of the abstract and discussion).

Reply: We changed the following content:

P2: Abstract

These findings suggest that the endogenous MOR system modulates altruism in the human brain.

P4: The current study

Altogether these data suggest that endogenous MOR system contributes to altruistic brain and its individual differences.

P19: Opioid system modulates brain activity during costly helping

However, given the behavioural pharmacological studies pointing towards causal effects of opiates on helping⁴⁵ our data contributes to a nascent literature associating the MOR system with brain process of prosocial decision-making.

Regarding both points it is important to mention the lack of a control condition and the intertwining of pain processing and donation-size in the limitation section, as well as that a specific design that can assess costly helping independently from the emotional response to seeing pain in others is needed to investigate the true relationship between neural activity and costly helping.

Reply: We added these in the limitations:

P20: Limitations

Another limitation of our experiment is the absence of a control condition. In future studies, incorporating a control condition would be beneficial to better distinguish the emotional responses associated with observing pain from those related to costly helping.

References

1. Karjalainen, T. *et al.* Dissociable roles of cerebral μ -opioid and type 2 dopamine receptors in vicarious pain: A combined PET-fMRI study. *Cereb. Cortex* **27**, 4257–4266 (2017).

2. Filkowski, M., Cochran, R. N. & Haas, B. Altruistic behavior: mapping responses in the brain. *Neurosci. Neuroeconomics* **Volume 5**, 65–75 (2016).
3. FeldmanHall, O. *et al.* Differential neural circuitry and self-interest in real vs hypothetical moral decisions. *Soc. Cogn. Affect. Neurosci.* **7**, 743–751 (2012).